# DERIVATIVE-FREE OPTIMIZATION VIA MONOTONIC STOCHASTIC SEARCH

## ABSTRACT

We consider the problem of minimizing a differentiable function $f : \mathbb{R}^d \to \mathbb{R}$ using only function evaluations, in the zeroth-order (derivative-free) setting. We propose three related monotone stochastic algorithms: the *Monotonic Stochastic Search* (MSS), persistent Monotonic Stochastic Search (pMSS), and MSS variant with gradient-approximation (MSSGA). MSS is a minimal stochastic direct-search method that samples a single Gaussian direction per iteration and performs an improve-or-stay update based on a single perturbation. For smooth non-convex objectives, we prove an averaged gradient-norm rate $\mathcal{O}(\sqrt{d}/\sqrt{T})$ in expectation, so that $\mathcal{O}(d/\varepsilon^2)$ function evaluations suffice to reach $\mathbb{E}\|\nabla f(\theta^t)\|_2 \leq \varepsilon$, improving the quadratic dependence on $d$ of deterministic direct search while matching the best known stochastic bounds. In addition, we propose a practical variant, pMSS, that reuses successful search directions with sufficient decrease, and establish that it guarantees $\liminf_{t\to\infty} \|\nabla f(\theta^t)\|_2 = 0$ almost surely. Since MSS relies solely on pairwise comparisons between $f(\theta^t)$ and $f(\theta^t + \alpha_t s_t)$, it falls within the class of optimization algorithms that assume access to an *exact* ranking oracle. We then generalize this framework to a *stochastic* ranking-oracle setting satisfying a local power-type margin condition, and demonstrate that a majority vote over $N$ noisy comparisons preserves the $\mathcal{O}(d/\varepsilon^2)$ gradient complexity in terms of iteration count, given suitably designed oracle queries. MSSGA uses finite-difference directional derivatives while enforcing monotonic descent. In the smooth non-convex regime, we show that the best gradient iterate satisfies $\min_{1\leq t\leq T} \|\nabla f(\theta^t)\|_2 = o(1/\sqrt{T})$ almost surely. To the best of our knowledge, this result provides the first $o(1/\sqrt{T})$ almost-sure convergence guarantee for gradient-approximation methods employing random directions. Furthermore, our analysis extends to the classical Random Gradient-Free (RGF) algorithm, establishing the same almost-sure convergence rate, which has not been previously shown for RGF. Finally, we show that MSS remains robust beyond the smooth setting: when $f$ is continuously differentiable, the iterates satisfy $\liminf_{t\to\infty} \|\nabla f(\theta^t)\|_2 = 0$ almost surely.

## 1 INTRODUCTION

We consider the problem of minimizing a function $f : \mathbb{R}^d \to \mathbb{R}$ in the absence of access to its derivatives, relying solely on a black-box oracle that provides function evaluations. The challenge is to minimize $f$ with as few oracle queries as possible. The methods used in this setting are called derivative-free (or zeroth-order) methods. They are crucial in many machine learning applications where computing gradients is impractical, expensive, or impossible. Examples include reinforcement learning (Malik et al., 2020; Mania et al., 2018; Salimans et al., 2017), black-box adversarial attacks on neural networks (Chen et al., 2017; Papernot et al., 2017; Ughi et al., 2022), hyperparameter tuning of deep networks (Turner et al., 2021; Koch et al., 2018; Snoek et al., 2012) and multi-agent target tracking (Al-Abri et al., 2021).

Two standard approaches have been proposed in the literature to address derivative-free optimization problems. The first involves estimating gradients using finite differences (Flaxman et al., 2005; Nesterov & Spokoiny, 2017). In (Nesterov & Spokoiny, 2017), it is shown that by estimating gradients with two function evaluations at nearby points, in the smooth non-convex, the smooth convex and the smooth strongly convex settings, one can obtain complexity bounds similar to those of traditional

gradient descent, but with an additional linear dependence on the dimensionality $d$ due to the cost of estimating gradients. This approach has been extended to the stochastic optimization setting, where the objective function is subject to randomness (Ghadimi & Lan, 2013; Duchi et al., 2015). Moreover, a variance reduction technique, inspired by gradient-based methods, was successfully adapted to the zeroth-order stochastic setting (Liu et al., 2018). The second approach to derivative-free optimization focuses on identifying a direction $s$ such that perturbing the current point along $s$ leads to an improvement in the objective function. Methods based on this idea are known as direct search methods. The search directions can be either deterministic (Vicente, 2013) or stochastic (Golovin et al., 2020; Bergou et al., 2020; Bakkali & Saadi, 2025). Deterministic direct search methods have been shown to achieve complexity bounds similar to traditional gradient descent, but with an additional quadratic dependence on the dimensionality (Vicente, 2013; Konečný & Richtárik, 2014). In contrast, (Bergou et al., 2020) presents a stochastic variant, the stochastic three-point (STP) method, which achieves a linear dependence on the dimension $d$ in the smooth non-convex setting, with a complexity bound of $\mathcal{O}(d/\epsilon^2)$, to obtain an $\epsilon$-stationary point in expectation. This improves upon the $\mathcal{O}(d^2/\epsilon^2)$ complexity of earlier deterministic direct search methods. Recently, in the smooth convex setting, the STP algorithm has been shown to maintain a linear dependence on the dimensionality, with complexity bound of $\mathcal{O}(d/\epsilon)$ (Bakkali & Saadi, 2025). However, this result requires that the objective function has a bounded sublevel set.

**Our Contribution** & **Related Work.** Our main contributions are:

- **A minimal monotone stochastic direct-search scheme (MSS).** We introduce Algorithm 1, which, at each iteration, samples a single Gaussian direction $s_t \sim \mathcal{N}(0, I_d)$ and sets $\theta^{t+1} = \operatorname{argmin}\{f(\theta^t), f(\theta^t + \alpha_t s_t)\}$. This can be seen as the natural stochastic extension of deterministic direct search (DDS) (Hooke & Jeeves, 1961; Kolda et al., 2003; Vicente, 2013), where the positive spanning set is replaced by a single random direction, and as a simplified version of GLD and STP: GLD (Golovin et al., 2020) samples many perturbations, possibly at several radii, and keeps the best, while STP (Bergou et al., 2020) evaluates $f$ at two symmetric points $x_t \pm \alpha_t s_t$. For smooth non-convex objectives, we prove that with stepsizes $\alpha_t = \alpha_0/\sqrt{dt}$ the averaged gradient norm satisfies $\frac{1}{T} \sum_{t=1}^{T} \mathbb{E}\big[\|\nabla f(\theta^t)\|_2\big] = \mathcal{O}\bigg(\sqrt{\frac{d}{T}}\bigg)$, so that $\mathcal{O}(d/\varepsilon^2)$ function evaluations suffice to reach $\mathbb{E}[\|\nabla f(\theta^t)\|_2] \leq \varepsilon$ (theorem 2). This improves upon the rate $\mathcal{O}(d^2/\varepsilon^2)$ obtained by DDS methods and achieving the best complexity bound for derivative free methods in the smooth non-convex setting. Although a comparable $\mathcal{O}\left(\sqrt{d/T}\right)$ rate for the best iterate was previously established for STP (Bergou et al., 2020), implying that the dependence on $d$ and $\varepsilon$ is not new for a stochastic direct search method, the following distinctions holds: i) MSS attains the convergence rate with a conceptually simpler improve or stay update based on a single perturbation and can be viewed as the natural stochastic extension of deterministic direct search. ii) The structure of MSS is also better adapted to comparison based feedback. A stochastic ranking oracle, when queried on a pair $(x, y)$, returns a random outcome whose bias is a function of the value difference $f(y) - f(x)$, that is, it provides noisy information about which of the two vectors has the smaller function value. MSS fits this interface exactly, since each iteration only requires comparisons between the current point $\theta^t$ and one perturbed point $\theta^t + \alpha_t s_t$, in contrast to STP which is built around decisions involving three points $\{\theta^t, \theta^t + \alpha_t s_t, \theta^t - \alpha_t s_t\}$. For this reason MSS is the natural building block in our stochastic ranking oracle extension (Section 2.3). iii) The single-direction design of MSS makes it especially amenable to a persistent variant that reuses successful directions. We therefore introduce a new algorithm, *persistent Monotonic Stochastic Search* (pMSS), a variant of MSS that reuses improving directions with sufficient decrease across iterations, thereby benefiting from a momentum-like effect that exploits successful improving directions, which is not the case for classical stochastic direct-search methods.

- **Comparison-based MSS with a stochastic ranking oracle.** We notice that MSS is inherently comparison-based and can operate without ever reading function values. In section 2.3 we analyze Algorithm 3, which only queries a stochastic ranking oracle returning noisy preferences between $\theta^t$ and $\theta^t + \alpha_t s_t$. Under a *local power-type margin* on the preference bias (Assumption 1), majority vote over $N$ comparisons yields a descent inequality that mirrors the exact-oracle case up to an additive $\mathcal{O}(N^{-1/(2p)})$ penalty (theorem 4). Choosing

$N$ polynomial in $1/\varepsilon$ recovers the same $\mathcal{O}(d/\varepsilon^2)$ gradient complexity in terms of iteration count as in the exact-value setting. This places our method within the landscape of preference-based optimization, but under significantly weaker structural assumption than Bradley–Terry–type models, where the function linking the preference probabilities and the function value differences must be known (Zhang & Ying, 2025). In particular, our analysis does not require knowledge of the link function, and we note that our *local power-type margin* assumption (Assumption 1) is both natural and, to the best of our knowledge, has not previously been used in the literature.

- **A persistent monotone scheme with sufficient decrease (pMSS).** Motivated by practice, we introduce a persistent variant of MSS, Algorithm 2, which reuses a downhill direction over several iterations whenever it produces a sufficient decrease. As in MSS, any non-increasing trial point $\theta^t + \beta_t s_t$ with $f(\theta^t + \beta_t s_t) \leq f(\theta^t)$ is accepted. Among such steps, pMSS distinguishes: (i) *sufficient-decrease* moves, where $f(\theta^t + \beta_t s_t) \leq f(\theta^t) - c\beta_t^2$ and the same pair $(s_t, \beta_t)$ is kept at the next iteration, and (ii) *marginal* moves, where $f(\theta^t) - c\beta_t^2 < f(\theta^t + \beta_t s_t) \leq f(\theta^t)$, in which case the algorithm still moves to the trial point but immediately resamples a fresh Gaussian direction and resets the stepsize from a deterministic sequence $\{a_k\}$; rejections also trigger resampling and a stepsize reset. This mechanism induces a simple momentum-like behavior: once a direction yields a streak of sufficient-decrease steps, the method advances along it for several iterations without additional randomness, while preserving global monotonicity of $f(\theta^t)$. We analyse pMSS via a block decomposition based on resampling times and show that, under smoothness and standard diminishing-stepsize conditions, the iterates satisfy $\liminf_{t\to\infty} \|\nabla f(\theta^t)\|_2 = 0$ almost surely (theorem 3). To the best of our knowledge, this is the convergence result for a random search method with persistent directions.

- **A monotone gradient-approximation scheme (MSSGA).** We introduce a variant of MSS and RGF, called MSSGA (Algorithm 4). This algorithm uses finite-difference directional derivatives in the spirit of Polyak's scheme (Polyak, 1987, Section 3.4) and Random Gradient-Free (RGF) methods (Nesterov & Spokoiny, 2017), but keeps the update *only* when it decreases $f$. Very recently, El Bakkali et al. (Bakkali & Saadi, 2025) obtained the first almost-sure convergence rates for *stochastic direct-search* methods in the smooth non-convex regime, showing that STP achieves $o(T^{-1/2+\epsilon})$ for any $\epsilon > 0$. In contrast, we prove that MSSGA enjoys the sharper almost-sure rate $o(1/\sqrt{T})$ for the best gradient iterate (theorem 6), matching the optimal $O(1/\sqrt{T})$ scaling known in expectation. To the best of our knowledge, this result provides the first $o(1/\sqrt{T})$ almost-sure convergence guarantee for gradient-approximation methods employing random directions. Moreover, our argument is not tied to the monotone acceptance rule and can be applied directly to the classical RGF algorithm, yielding the same $o(1/\sqrt{T})$ almost-sure rate in the smooth non-convex case.

- **Non-smooth analysis for monotone stochastic direct search.** Finally, we study MSS in a non-smooth regime where $f$ is only assumed to be continuously differentiable and the initial sublevel set is bounded (Assumption 2). When the search directions are drawn uniformly from the sphere, and the stepsizes satisfy $\alpha_t \to 0$ and $\sum_t \alpha_t = \infty$, we show that $\liminf_{t\to\infty} \|\nabla f(\theta^t)\|_2 = 0$ almost surely, hence the trajectory admits accumulation points that are stationary (theorem 7). This extends the almost-sure stationarity theory for stochastic direct-search methods beyond the smooth setting.

## 2 CONVERGENCE ANALYSIS FOR THE CLASS OF SMOOTH NON-CONVEX FUNCTIONS

### 2.1 CONVERGENCE ANALYSIS FOR MSS ALGORITHM

In this subsection, we focus on the monotonic stochastic search algorithm, which is presented below:

---

**Algorithm 1** Monotonic Stochastic Search algorithm (MSS)

---

1: **Input:**
2:   $\theta^1 \in \mathbb{R}^d$: Initial parameter vector
3:   $\{\alpha_t\}_{t \geq 1}$: Step-size sequence
4: **for** $t = 1, 2, \ldots$ **do**
5:     Sample search direction: $s_t \sim \mathcal{N}(0, I_d)$
6:     $\theta^{t+1} = \operatorname{argmin}_{\theta \in \{\theta^t, \theta^t + \alpha_t s_t\}} f(\theta)$
7: **end for**

---

**Lemma 1.** *Let $\{\theta^t\}_{t \geq 1}$ be a sequence generated by algorithm 1. Assuming that $f$ is $L-$smooth, the following inequality holds for all $t \geq 1$:*

$$\frac{1}{\sqrt{2\pi}} \alpha_t \mathbb{E}\left[\left\|\nabla f\left(\theta^t\right)\right\|_2\right] \leq \mathbb{E}\left[f\left(\theta^t\right)\right] - \mathbb{E}\left[f\left(\theta^{t+1}\right)\right] + \frac{Ld\alpha_t^2}{4}.$$

The inequality in lemma 1 is analogous to the key inequality used in the analysis of gradient descent (GD) for smooth functions. In the standard GD setting, where updates are given by $\theta^{t+1} = \theta^t - \frac{1}{L}\nabla f(\theta^t)$, with $L$ as the smoothness parameter of $f$, the smoothness of $f$ ensures the following descent property: $\frac{1}{2L}\|\nabla f(\theta^t)\|_2^2 \leq f(\theta^t) - f(\theta^{t+1})$. This inequality ensures that the averaged gradient norm iterate generated by GD algorithm, converges to zero at a rate of $\mathcal{O}(1/\sqrt{T})$. Similarly, lemma 1 provides an analogous inequality tailored to our algorithm, which ensures a similar convergence rate for the averaged gradient iterate produced by the MSS algorithm. Specifically, we show that under the MSS algorithm, the averaged gradient iterate converges in expectation to zero at a rate of $\mathcal{O}(\sqrt{d}/\sqrt{T})$. It's worth noting that for GD method, the convergence rate is independent of the dimensionality of the space.

**Constant step size.** By averaging the inequality of lemma 1 over the iterations 1 to $T$, while using a constant step size $\alpha_t = \frac{\alpha_0}{\sqrt{dT}}$, we obtain the following theorem.

**Theorem 1.** *Assume that $f$ is $L-$smooth and lower and let $T \geq 1$. By following algorithm 1 using the constant step size $\alpha_t = \frac{\alpha_0}{\sqrt{dT}}$, we obtain:*

$$\frac{\sum_{t=1}^T \mathbb{E}[\|\nabla f(\theta^t)\|_2]}{T} \leq \left(\frac{\sqrt{2\pi}\left(f(\theta^1) - \inf_{\theta \in \mathbb{R}^d} f(\theta)\right)}{\alpha_0} + \frac{\sqrt{\pi}\alpha_0 L}{2\sqrt{2}}\right)\sqrt{\frac{d}{T}}.$$

Theorem 1 implies that for a fixed number of iterations $T$, by choosing constant step sizes dependent on $T$, the average $\frac{\sum_{t=1}^T \mathbb{E}[\|\nabla f(\theta^t)\|_2]}{T}$, and subsequently the best iterate $\min_{1 \leq t \leq T} \mathbb{E}[\|\nabla f(\theta^t)\|_2]$, can be bounded above by an accuracy of order $\mathcal{O}(\sqrt{\frac{d}{T}})$. However, if we aim for a precision of order $\frac{\sqrt{d}}{\sqrt{T'}}$ with $T' > T$, we must restart the iterations with a new step size dependent on $T'$. We note also that Theorem 1 does not imply that the best gradient iterate converges to zero, since the step sizes are tied to a fixed accuracy. In the next theorem, we show that by choosing diminishing step sizes $\alpha_t = \frac{\alpha_0}{\sqrt{dt}}$, convergence is guaranteed.

**Diminishing step sizes.** We show that if $\{\theta^t\}_{t \geq 1}$ is generated by algorithm 1 with step sizes $\alpha_t = \alpha_0/\sqrt{dt}$ with $\alpha_0 > 0$, then the averaged gradient norm iterate $(1/T)\sum_{t=1}^T \|\nabla f(\theta^t)\|_2$ converges in expectation to zero at a rate of $\mathcal{O}\left(\sqrt{d}/\sqrt{T}\right)$. This result is stated in theorem 2.

**Theorem 2.** *Let $\{\theta^t\}_{t \geq 1}$ be a sequence generated by algorithm 1 with step sizes $\alpha_t = \frac{\alpha_0}{\sqrt{dt}}$ for $\alpha_0 > 0$. Assuming that $f$ is $L-$smooth and lower bounded, the following inequality holds for all $T \geq 2$:*

$$\frac{\sum_{t=1}^T \mathbb{E}\left[\|\nabla f(\theta^t)\|_2\right]}{T} \leq \left(\frac{2\sqrt{2\pi}\left(f(\theta^1) - \inf_{\theta \in \mathbb{R}^d} f(\theta)\right)}{\alpha_0} + \sqrt{\frac{\pi}{2}}L\alpha_0\right)\sqrt{\frac{d}{T}}.$$

**Remark 1.** *The bound in theorem 2 shows that the best gradient iterate $\min_{1 \leq t \leq T} \|\nabla f(\theta^t)\|_2$ also converges to zero in expectation, at a rate of $\mathcal{O}\left(\sqrt{\frac{d}{T}}\right)$. A similar convergence rate can be established almost surely. Indeed, given $\epsilon \in (0, \frac{1}{2})$, by applying lemma 1 with step size sequence $\{\alpha_t\}_{t \geq 1}$ defined by $\alpha_t = \frac{1}{t^{\frac{1}{2}+\epsilon}}$, it follows that $\mathbb{E}\left[\sum_{T=1}^{\infty} \frac{1}{T^{\frac{1}{2}+\epsilon}} \min_{1 \leq t \leq T} \|\nabla f(\theta^t)\|_2\right] < \infty$, and consequently, $\sum_{T=1}^{\infty} \frac{1}{T^{\frac{1}{2}+\epsilon}} \min_{1 \leq t \leq T} \|\nabla f(\theta^t)\|_2 < \infty$ a.s. We also have $\lim_{T \to \infty} \min_{1 \leq t \leq T} \|\nabla f(\theta^t)\|_2 = 0$ which follows as a consequence of theorem 2. Applying (Bakkali & Saadi, 2025, Lemma 5), we conclude that $\min_{1 \leq t \leq T} \|\nabla f(\theta^t)\|_2 = o\left(1/\left(\sum_{t=1}^{T} \frac{1}{t^{\frac{1}{2}+\epsilon}}\right)\right)$, and since $\sum_{t=1}^{T} \frac{1}{t^{\frac{1}{2}+\epsilon}} \sim T^{\frac{1}{2}-\epsilon}$, it follows that $\min_{1 \leq t \leq T} \|\nabla f(\theta^t)\|_2 = o\left(\frac{1}{T^{\frac{1}{2}-\epsilon}}\right)$ almost surely. We include this for completeness, as the proof follows directly from (Bakkali & Saadi, 2025, Lemma 5) and the inequality given in lemma 1.*

## 2.2 CONVERGENCE ANALYSIS FOR pMSS ALGORITHM

In this section we introduce the persistent Monotonic Stochastic Search algorithm (pMSS), a practical variant of MSS. As in MSS, any non-increasing trial point $\theta^t + \beta_t s_t$ with $f(\theta^t + \beta_t s_t) \leq f(\theta^t)$ is accepted. pMSS then distinguishes two types of accepted steps: if the decrease is *sufficient*, $f(\theta^t + \beta_t s_t) \leq f(\theta^t) - c\beta_t^2$, it *persists* by reusing the same direction and step-size at the next iteration; if the decrease is only marginal, i.e., $f(\theta^t) - c\beta_t^2 < f(\theta^t + \beta_t s_t) \leq f(\theta^t)$, it still moves to the trial point but immediately resamples a new Gaussian direction and resets the step-size from a fixed step-size sequence $\{a_k\}_{k \geq 1}$. When the trial point is worse, $f(\theta^t + \beta_t s_t) > f(\theta^t)$, pMSS rejects it and also resamples. This persistence mechanism lets pMSS chain several steps along particularly good directions while preserving monotonicity.

---

**Algorithm 2** Persistent MSS with sufficient decrease (pMSS)

---

1: **Inputs:** initial $\theta^1 \in \mathbb{R}^d$; stepsizes $\{a_k\}_{k \geq 1} \subset (0, \infty)$; margin $c > 0$.
2: $k = 1$; draw $s_1 \sim \mathcal{N}(0, I_d)$; set $\beta_1 = a_1$.
3: **for** $t = 1, 2, \dots$ **do**
4:  **if** $f(\theta^t + \beta_t s_t) \leq f(\theta^t)$ **then**                  ▷ decrease (nonincrease) step
5:    $\theta^{t+1} = \theta^t + \beta_t s_t$                                   ▷ accept
6:    **if** $f(\theta^t + \beta_t s_t) \leq f(\theta^t) - c\beta_t^2$ **then**   ▷ sufficient decrease
7:      $s_{t+1} = s_t, \beta_{t+1} = \beta_t$                                ▷ accept and persist
8:    **else**
9:      draw $s_{t+1} \sim \mathcal{N}(0, I_d)$; $k = k + 1$; set $\beta_{t+1} = a_k$   ▷ accept but do not persist
10:    **end if**
11:  **else**
12:    $\theta^{t+1} = \theta^t$                                             ▷ reject
13:    draw $s_{t+1} \sim \mathcal{N}(0, I_d)$; $k = k + 1$; set $\beta_{t+1} = a_k$   ▷ reject and resample
14:  **end if**
15: **end for**

---

Let $(\Omega, \mathcal{F}, \mathbb{P})$ be a probability space on which all random variables $\{\theta^t, \beta_t, s_t\}_{t \geq 1}$ are defined. The "pre-direction" information at each $t$ is given by: $\mathcal{F}_t := \sigma\left(\theta^1, \beta_1, s_1, \dots, \theta^{t-1}, \beta_{t-1}, s_{t-1}, \theta^t, \beta_t\right)$. For each $t \geq 1$ define the sufficient-decrease event $A_t := \left\{f(\theta^t + \beta_t s_t) \leq f(\theta^t) - c\beta_t^2\right\}$, and its complement $R_t := A_t^{\complement}$, which corresponds to the case where there is no sufficient decrease (this includes both marginal accepts and rejections). In the algorithm, whenever $R_t$ occurs we draw a new Gaussian direction at time $t + 1$ and reset the stepsize from the sequence $\{a_k\}_{k \geq 1}$.

**Resampling times.** We now formalize the times $\tau_k$ at which a fresh direction is drawn. Set $\tau_1 := 1$. Recursively, define

$$\rho_k := \begin{cases} 0, & \text{if } A_{\tau_k} \text{ does not occur,} \\ \sup\left\{m \geq 1 : A_{\tau_k}, A_{\tau_k+1}, \dots, A_{\tau_k+m-1} \text{ all occur}\right\}, & \text{if } A_{\tau_k} \text{ occurs,} \end{cases}$$

and then set $\tau_{k+1} := \tau_k + \rho_k + 1$, with the convention that $\tau_{k+1} := \infty$ if $\rho_k = \infty$.

Case $\rho_k = 0$ (no sufficient decrease after resampling)

$$s_{\tau_{k+1}},\ \beta_{\tau_{k+1}} = a_{k+1}$$

$\tau_k$    $\tau_{k+1}$
$= \tau_k + 1$    $\longrightarrow t$

Case $0 < \rho_k < \infty$ (block with sufficient decrease)

$A_{\tau_k}, \ldots, A_{\tau_k + \rho_k - 1}$ hold

$R_{\tau_k + \rho_k}$ holds    $s_{\tau_{k+1}},\ \beta_{\tau_{k+1}} = a_{k+1}$

$\tau_k$    $\tau_k + 1$    $\cdots$    $\tau_k + \rho_k - 1$    $\tau_k + \rho_k$    $\tau_{k+1}$    $\longrightarrow t$

$$s_t = s_{\tau_k},\ \beta_t = a_k,\quad \forall t \in \{\tau_k, \ldots, \tau_k + \rho_k\}$$

Figure 1: Resampling times $\tau_k$ and block lengths $\rho_k$ in pMSS.

Lemma 4 implies that resampling occurs infinitely many times almost surely. At each such resampling time we draw a fresh Gaussian search direction, independent of the past. Therefore, by repeating the proof of Lemma 1 at these resampling times, we obtain the following result.

**Theorem 3.** *Assume that $f$ is $L-$smooth, lower bounded, and that the stepsizes $\{a_k\}_{k \geq 1}$ in Algorithm 2 satisfy: $\sum_{k=1}^{\infty} a_k = \infty$ and $\sum_{k=1}^{\infty} a_k^2 < \infty$. Let $\{\theta^t\}_{t \geq 1}$ be generated by Algorithm 2. We have:*
$$\liminf_{t \to \infty} \left\|\nabla f(\theta^t)\right\|_2 = 0 \quad almost\ surely.$$

**Discussion and illustrative example.** The persistence mechanism in pMSS is particularly advantageous for objectives where certain directions admit long sequences of successful steps with sufficient decrease. Once such favorable directions are identified, pMSS is able to repeatedly exploit them, in contrast to stochastic search methods, like STP, that must restart their exploration at each iteration. This repeated isotropic exploration is costly, due to its linear dependence on the dimension. Consequently, pMSS attains a more favorable exploration–exploitation trade-off. We illustrate this first on a two-dimensional quadratic function $f(x) = \frac{1}{2}\left(x_1^2 + 10^{-2}x_2^2\right) + x_1 - 0.2\,x_2$, whose level sets form a long valley aligned with the $x_2$-axis. Starting from the same point, STP keeps resampling directions and therefore zigzags across the valley, making only small net progress toward the minimizer. In contrast, once pMSS hits a sufficiently good direction, it keeps reusing it and advances almost straight along the valley floor.

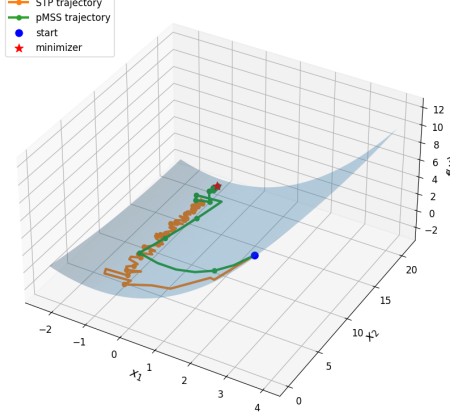

Figure 2: STP and pMSS trajectories on the 2D valley quadratic.

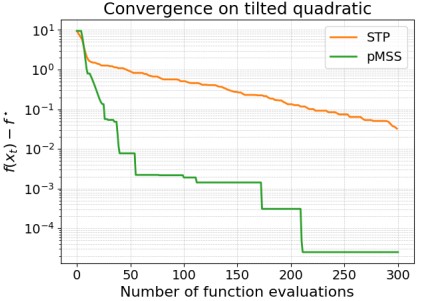

Figure 3: Function gap $f(\theta^t) - f^\star$ versus function evaluations on the same 2D problem.

We now embed the same valley in higher dimensions by adding orthogonal directions with unit curvature, $f(x) = \frac{1}{2}\left(x_1^2 + 10^{-2}x_2^2 + \sum_{i=3}^{d} x_i^2\right) + x_1 - 0.2\,x_2$. Figure 4 compares STP and pMSS on this function for $d \in \{100, 300, 500, 1000\}$ under a common budget of function evaluations. As the dimension grows, the advantage of pMSS over STP increases with it.

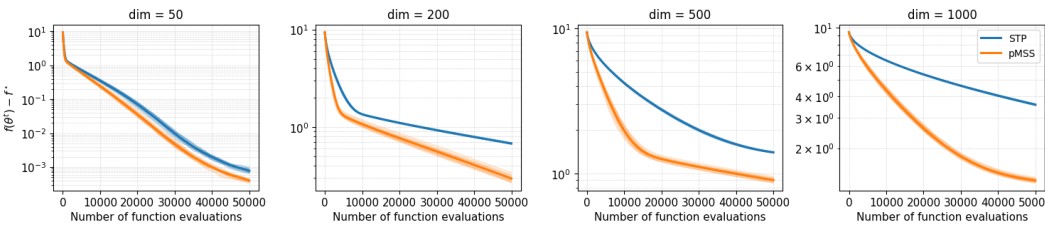

Figure 4: Diagonal valley quadratic in dimensions $d \in \{100, 300, 500, 1000\}$: function gap $f(\theta^t) - f^\star$ versus number of function evaluations, averaged over 20 runs.

## 2.3 Convergence analysis for MSS algorithm with stochastic ranking oracle

We remark that MSS is inherently comparison-based and does not require reading function values. The analysis in section 2.1 assumed an exact (noise-free) comparator. We now relax this to a stochastic ranking oracle that returns noisy preferences but is biased toward the correct ordering with a local power-type advantage in the function-value gap. We formalize this in the following assumption.

**Assumption 1.** *For any pair $(\theta^1, \theta^2) \in \mathbb{R}^d \times \mathbb{R}^d$, the oracle returns a single outcome $o \in \{0, 1\}$ interpreted as "$\theta^2$ is better" ($o = 1$) or "$\theta^1$ is better" ($o = 0$), with bias toward the truly better point:*

$$\begin{aligned}
\text{if } f(\theta^2) < f(\theta^1): \quad &\mathbb{P}(o = 1) \geq \tfrac{1}{2} + h\big(|f(\theta^2) - f(\theta^1)|\big), \\
\text{if } f(\theta^1) < f(\theta^2): \quad &\mathbb{P}(o = 0) \geq \tfrac{1}{2} + h\big(|f(\theta^2) - f(\theta^1)|\big), \\
\text{if } f(\theta^1) = f(\theta^2): \quad &\mathbb{P}(o = 1) = \tfrac{1}{2}.
\end{aligned}$$

*Here $h : \mathbb{R}_+ \to [0, \frac{1}{2}]$ is nondecreasing (not necessarily continuous), $h(0) = 0$, and there exist $r > 0$, $\kappa > 0$, and $p \geq 1$ such that $\forall x \in [0, r]$, $h(x) \geq \kappa\, x^p$. We denote $m_r := h(r) > 0$.*

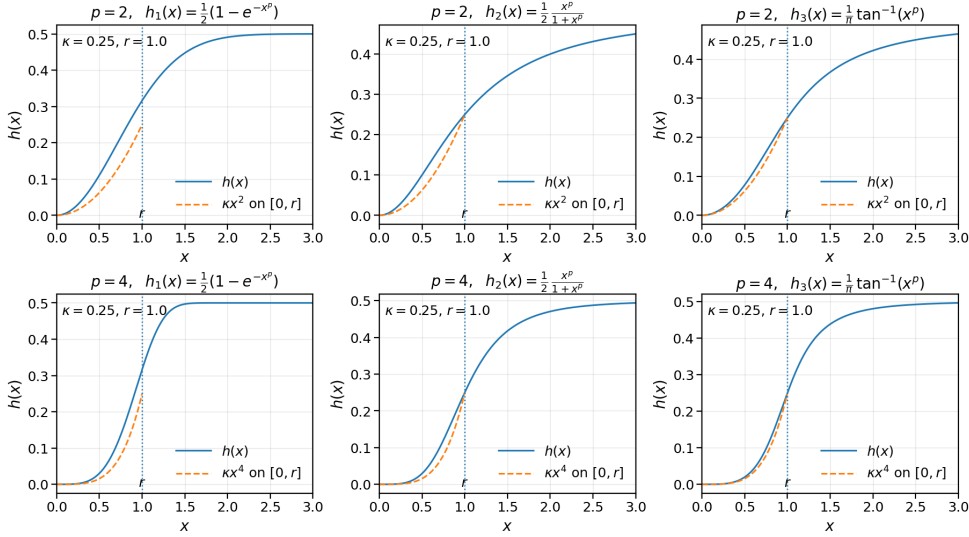

Figure 5: Examples of admissible margin functions.

**Algorithm.** At iteration $t$, sample $s_t \sim \mathcal{N}(0, I_d)$, form the candidate $\theta^t + \alpha_t s_t$, collect $N$ i.i.d. oracle outcomes on the pair, aggregate by majority, and accept if the majority prefers the candidate.

---

**Algorithm 3** MSS with Stochastic Ranking Oracle

---

1: **Input:** initial point $\theta^1 \in \mathbb{R}^d$, stepsizes $(\alpha_t)$, comparisons $N$
2: **for** $t = 1, 2, \ldots$ **do**
3:     Sample $s_t \sim \mathcal{N}(0, I_d)$
4:     Query oracle $N$ times on $(\theta^t, \theta^t + \alpha_t s_t)$, get $o_{t,1}, \ldots, o_{t,N} \in \{0, 1\}$
5:     $\bar{o}_t = \dfrac{1}{N} \sum_{n=1}^{N} \mathbf{1}\{o_{t,n} = 1\}$
6:     $\theta^{t+1} \leftarrow \begin{cases} \theta^t + \alpha_t s_t, & \bar{o}_t > \frac{1}{2}, \\ \theta^t, & \text{otherwise.} \end{cases}$
7: **end for**

---

**Notation and decision events.** Let

$$\Delta_t := f(\theta^t + \alpha_t s_t) - f(\theta^t), \quad A_t := \{\bar{o}_t > \tfrac{1}{2}\}, \quad B_t := \{f(\theta^t) > f(\theta^t + \alpha_t s_t)\}.$$

Since the update is accept–or–stay, $f(\theta^{t+1}) = f(\theta^t) + \Delta_t \mathbf{1}_{A_t}$, to derive a descent inequality comparable to the standard MSS bound in lemma 1 we control $\mathbb{E}[\Delta_t \mathbf{1}_{A_t} \mid \theta^t]$ via the elementary split $\Delta_t \mathbf{1}_{A_t} = \underbrace{\Delta_t \mathbf{1}_{B_t}}_{\text{true-improvement term}} + \underbrace{\Delta_t (\mathbf{1}_{A_t} - \mathbf{1}_{B_t})}_{\text{ranking-error term}}$. The first term coincides with the exact comparator case and can be upper bounded using $L$-smoothness and Gaussian symmetry, which yields a linear decrease in $\|\nabla f(\theta^t)\|_2$ up to an $O(d\alpha_t^2)$ term, see Lemma 5. The second term captures ranking mistakes. Under assumption 1, the majority vote error probability after $N$ comparisons decays as $\exp(-2Nh(|\Delta_t|)^2)$, which leads to two regimes: for small gaps, $|\Delta_t| \leq r$, the local power margin $h(x) \geq \kappa x^p$ controls the contribution of this term by $O(N^{-1/(2p)})$, see Lemma 6 and Lemma 7; for larger gaps, $|\Delta_t| \geq r$, the noise penalty is exponentially small in $N$ and only rescales the exact MSS bound by a factor $e^{-2Nm_r^2}$ with $m_r = h(r)$. Putting these ingredients together gives a descent inequality of the form in Lemma 8, where the exact comparator bound from Lemma 1 is recovered up to constants and an additional noise term of order $N^{-1/(2p)}$.

---

**Theorem 4.** *If* $N \geq \left\lceil \dfrac{\ln 4}{2m_r^2} \right\rceil$, *then* $e^{-2Nm_r^2} \leq \frac{1}{4}$ *and, for all* $t \geq 1$,

$$\frac{\alpha_t}{2\sqrt{2\pi}} \, \mathbb{E}\big[\|\nabla f(\theta^t)\|_2\big] \; \leq \; \mathbb{E}\big[f(\theta^t) - f(\theta^{t+1})\big] + \frac{5}{8} \, L \, d \, \alpha_t^2 + \frac{1}{(4ep\kappa^2 N)^{\frac{1}{2p}}}.$$

---

$\varepsilon$**-complexity.** Let $f_\star := \inf_\theta f(\theta)$ and $\Delta_f := f(\theta^1) - f_\star < \infty$. Fix $\varepsilon \in (0, 1]$ and take a constant stepsize $\alpha_t := \frac{4\,\varepsilon}{15\,\sqrt{2\pi}\,L\,d}$. If, in addition, $N \geq N_\varepsilon := \max\left( \left\lceil \frac{\ln 4}{2m_r^2} \right\rceil, \left\lceil \frac{1}{4ep\kappa^2} \left( \frac{45\,\pi\,L\,d}{\varepsilon^2} \right)^{2p} \right\rceil \right)$, then after $T_\varepsilon := \left\lceil \frac{45\,\pi\,L\,d\,\Delta_f}{\varepsilon^2} \right\rceil$ iterations we have $\min_{1 \leq t \leq T_\varepsilon} \mathbb{E}\big[\|\nabla f(\theta^t)\|_2\big] \leq \varepsilon$.

## 2.4 CONVERGENCE ANALYSIS FOR MSS ALGORITHM WITH GRADIENT APPROXIMATION

In this subsection, we focus on the monotonic stochastic search algorithm with gradient approximation, which is presented below:

---

**Algorithm 4** Monotonic Stochastic Search algorithm with Gradient Approximation (MSSGA)

---

1: **Input:**
2: $\theta^1 \in \mathbb{R}^d$: Initial parameter vector
3: $\alpha > 0$: Step size parameter
4: $\{\gamma_t\}_{t \geq 1}$: Sequence of smoothing parameters
5: **for** $t = 1, 2, \ldots$ **do**
6:  Sample a search direction $s_t$ uniformly from the unit sphere $\mathbb{S}^{d-1} = \{v \in \mathbb{R}^d : \|v\|_2 = 1\}$
7:  Update the current vector:

$$\theta^{t+1} = \mathrm{argmin}_{\theta \in \{\theta^t, \theta^t - \alpha \frac{f(\theta^t + \gamma_t s_t) - f(\theta^t)}{\gamma_t} s_t\}} f(\theta)$$

8: **end for**

---

We show that if $\{\theta^t\}_{t \geq 1}$ is generated by this algorithm with a step size parameter $\alpha \leq \frac{1}{L}$, then the averaged squared gradient norm, $\frac{1}{T} \sum_{t=1}^T \|\nabla f(\theta^t)\|_2^2$, converges in expectation to zero at a rate of $\mathcal{O}\left(\frac{d}{T}\right)$. This result follows directly from theorem 5.

It is important to note that algorithm 4 is not a special case of algorithm 1, as it does not rely on a predetermined step size sequence—the step sizes are instead chosen adaptively.

**Lemma 2.** *Assume that $f$ is $L-$smooth and let $\{\theta^t\}_{t \geq 1}$ be a sequence generated by algorithm 4 with $\alpha \leq \frac{1}{L}$. We have the following inequality for all $t \geq 1$:*

$$\mathbb{E}[\|\nabla f(\theta^t)\|_2^2] \leq \frac{2d(\mathbb{E}[f(\theta^t)] - \mathbb{E}[f(\theta^{t+1})])}{\alpha} + \frac{dL^2}{4}\gamma_t^2.$$

**Remark 2.** *If the sequence of smoothing parameters satisfies $\sum_{t=1}^\infty \gamma_t^2 < \infty$, then lemma 2 implies that $\sum_{t=1}^\infty \mathbb{E}[\|\nabla f(\theta^t)\|_2^2] < \infty$, which in turn implies that $\lim_{t \to \infty} \mathbb{E}\left[\|\nabla f(\theta^t)\|_2^2\right] = 0$. By Cauchy–Schwarz inequality, we can then deduce that the gradient norm at the last iterate, $\|\nabla f(\theta^t)\|_2$, converges to zero in expectation.*

By averaging the sides of the inequality in lemma 2, we obtain the inequality in theorem 5.

**Theorem 5.** *Assume that $f$ is $L-$smooth, lower bounded and let $\{\theta^t\}_{t \geq 1}$ be a sequence generated by algorithm 4 with $\alpha = \frac{1}{L}$. For all $T \geq 1$, we have:*

$$\frac{\sum_{t=1}^T \mathbb{E}[\|\nabla f(\theta^t)\|_2^2]}{T} \leq \frac{2dL(f(\theta^1) - \inf_{\theta \in \mathbb{R}^d} f(\theta))}{T} + \frac{dL^2}{4}\frac{\sum_{t=1}^T \gamma_t^2}{T}.$$

*In particular, if $\sum_{t=1}^\infty \gamma_t^2 < \infty$, we obtain a complexity bound of $\mathcal{O}\left(\frac{d}{T}\right)$.*

**Remark 3.** *Given that $\sum_{t=1}^\infty \gamma_t^2 < \infty$, the Cauchy–Schwarz inequality together with theorem 5 implies that $\frac{\sum_{t=1}^T \mathbb{E}[\|\nabla f(\theta^t)\|_2]}{T} = \mathcal{O}\left(\frac{\sqrt{d}}{\sqrt{T}}\right)$. This bound shows that both the average gradient iterate and the best gradient iterate converge to zero in expectation, at a rate of $\mathcal{O}\left(\frac{\sqrt{d}}{\sqrt{T}}\right)$.*

We now establish that the convergence rate achieved for the best gradient iterate in expectation is also attained almost surely, as detailed in theorem 6.

**Theorem 6.** *Assume that $f$ is $L-$smooth, lower bounded and let $\{\theta^t\}_{t \geq 1}$ be a sequence generated by algorithm 4 with $\alpha \leq \frac{1}{L}$ and $\sum_{t=1}^\infty \gamma_t^2 < \infty$. We have:*

$$\min_{1 \leq t \leq T} \|\nabla f(\theta^t)\|_2 = o\left(\frac{1}{\sqrt{T}}\right) \quad \text{almost surely.}$$

### 2.5 CONVERGENCE ANALYSIS FOR MSS ALGORITHM IN THE NON-SMOOTH SETTING

In this section, we analyze the convergence of algorithm 1 when applied to non-smooth objectives, using a uniform distribution over the unit sphere (instead of a Gaussian) to simplify the analysis. Although $f$ may lack smoothness (e.g., the gradient may not be Lipschitz), we work under the following assumption.

**Assumption 2.** $f$ *is continuously differentiable, lower bounded, and the level set* $\mathcal{L}(\theta^1) := \{\theta : f(\theta) \leq f(\theta^1)\}$ *is bounded.*

**Good directions.** For any $\theta \in \mathbb{R}^d$, define the set of "good" directions:

$$A_d(\theta) := \left\{ s \in \mathbb{S}^{d-1} : \langle \nabla f(\theta), s \rangle \leq -\frac{1}{2\sqrt{d}} \|\nabla f(\theta)\|_2 \right\}.$$

Intuitively, $A_d(\theta)$ contains directions that are sufficiently aligned with $-\nabla f(\theta)$, ensuring a uniform amount of decrease in function value whenever the gradient is non-negligible.

---

**Theorem 7.** *Assume that assumption 2 holds, and let* $\{\theta^t\}$ *be the sequence generated by algorithm 1 when the search directions* $s_t$ *are drawn uniformly from the unit sphere. Suppose that* $\alpha_t > 0$ *for all* $t$, $\alpha_t \to 0$, *and* $\sum_{t=1}^{\infty} \alpha_t = +\infty$. *Then*

$$\liminf_{t \to \infty} \|\nabla f(\theta^t)\|_2 = 0 \qquad \textit{almost surely.}$$

---

The proof, given in the appendix, combines the constant probability of sampling a good direction (Lemma 9) with a uniform descent property (Lemma 10) to obtain an expected decrease inequality (Lemma 11). A standard Robbins–Siegmund argument then yields theorem 7.

## 3 EXPERIMENTS

We evaluate MSS and pMSS in a policy-search setting with pairwise preference feedback, following the Zeroth-Order Policy Gradient (ZPG) framework (Zhang & Ying, 2025). For two policies $\pi_\theta, \pi_{\theta'}$ with returns $R(\pi_\theta)$ and $R(\pi_{\theta'})$, a synthetic preference is drawn as $\pi_{\theta'} \succ \pi_\theta \iff$ Bernoulli$\big(\sigma(R(\pi_{\theta'}) - R(\pi_\theta))\big) = 1$, where $\sigma(t) = 1/(1 + e^{-t})$ is the logistic link. Each policy evaluation uses $N = 64$ trajectories, aggregated into a single Bernoulli comparison ($M = 1$).

We consider CartPole-v1, InvertedPendulum-v5, and Swimmer-v5 from Gymnasium, with the same neural policy architecture for all methods: a two-layer MLP with 64 hidden units per layer and $\tanh$ activations. ZPG uses a random-direction finite-difference estimator with smoothing parameter $\mu = 10^{-2}$ and stepsize $\alpha = 10^{-3}$. MSS and pMSS use a constant stepsize $\alpha = 10^{-1}$. In pMSS, a candidate policy is accepted and the current search direction is reused only if the estimated probability that it is better than the incumbent exceeds $0.7$ (probability margin $0.7$); otherwise a new random direction is sampled. We run each method for a fixed budget of policy evaluations and report the mean return over 10 seeds; shaded regions in Figure 6 show one standard deviation.

Figure 6 compares MSS, pMSS, and ZPG. On CartPole-v1, pMSS learns fastest and reaches near-maximum reward, with MSS catching up later and ZPG clearly lagging behind. These results indicate that monotonic stochastic search in policy space is at least competitive with, and often superior to, zeroth-order gradient estimation in this preference-based setting.

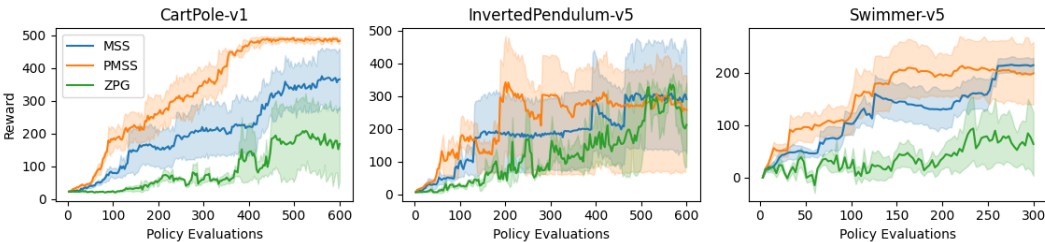

Figure 6: Average return vs. evaluations on three control tasks.

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

# A  APPENDIX

## A.1  CONVERGENCE ANALYSIS FOR THE CLASS OF SMOOTH NON-CONVEX FUNCTIONS

### A.1.1  CONVERGENCE ANALYSIS FOR MSS ALGORITHM

*Proof of Lemma 1.* Let $t \geq 1$. Since $f$ is $L$-smooth, we have:

$$f(\theta^t + \alpha_t s_t) \leq f(\theta^t) + \alpha_t \langle \nabla f(\theta^t), s_t \rangle + \frac{L}{2} \alpha_t^2 \|s_t\|_2^2.$$

Rearranging the terms and using the fact that $f(\theta^{t+1}) \leq f(\theta^t + \alpha_t s_t)$, we obtain:

$$-\alpha_t \langle \nabla f(\theta^t), s_t \rangle \leq f(\theta^t) - f(\theta^{t+1}) + \frac{L}{2} \alpha_t^2 \|s_t\|_2^2.$$

By multiplying by $\mathbf{1}_{\{\langle \nabla f(\theta^t), s_t \rangle \leq 0\}}$, we obtain:

$$\alpha_t |\langle \nabla f(\theta^t), s_t \rangle| \mathbf{1}_{\langle \nabla f(\theta^t), s_t \rangle \leq 0} \leq \underbrace{f(\theta^t) - f(\theta^{t+1})}_{\geq 0} + \frac{L}{2} \alpha_t^2 \|s_t\|_2^2 \mathbf{1}_{\{\langle \nabla f(\theta^t), s_t \rangle \leq 0\}}.$$

This implies that:

$$\alpha_t |\langle \nabla f(\theta^t), s_t \rangle| \mathbf{1}_{\{\langle \nabla f(\theta^t), s_t \rangle \leq 0\}} \leq f(\theta^t) - f(\theta^{t+1}) + \frac{L}{2} \alpha_t^2 \|s_t\|_2^2 \mathbf{1}_{\{\langle \nabla f(\theta^t), s_t \rangle \leq 0\}} \mathbf{1}_{\{\nabla f(\theta^t) \neq 0\}} \quad (1)$$

Since the PDF of the distribution $\mathcal{N}(0, I_d)$ is origin-symmetric, we obtain:

$$\mathbb{E}[|\langle \nabla f(\theta^t), s_t \rangle| \mathbf{1}_{\{\langle \nabla f(\theta^t), s_t \rangle \geq 0\}} \mid \theta^t] = \mathbb{E}[|\langle \nabla f(\theta^t), -s_t \rangle| \mathbf{1}_{\{\langle \nabla f(\theta^t), -s_t \rangle \geq 0\}} \mid \theta^t]$$
$$= \mathbb{E}[|\langle \nabla f(\theta^t), s_t \rangle| \mathbf{1}_{\{\langle \nabla f(\theta^t), s_t \rangle \leq 0\}} \mid \theta^t].$$

Since $\mathbf{1}_{\{\langle \nabla f(\theta^t), s_t \rangle \leq 0\}} + \mathbf{1}_{\{\langle \nabla f(\theta^t), s_t \rangle \geq 0\}} \geq 1$, it holds that:

$$\mathbb{E}[|\langle \nabla f(\theta^t), s_t \rangle| \mathbf{1}_{\{\langle \nabla f(\theta^t), s_t \rangle \leq 0\}} \mid \theta^t] \geq \frac{1}{2} \mathbb{E}[|\langle \nabla f(\theta^t), s_t \rangle| \mid \theta^t].$$

Combining this with eq. (1), we get:

$$\frac{\alpha_t}{2}\mathbb{E}[|\langle\nabla f(\theta^t), s_t\rangle| \mid \theta^t] \leq \mathbb{E}[f(\theta^t)-f(\theta^{t+1}) \mid \theta^t]+\frac{L\alpha_t^2}{2}\mathbb{E}\big[||s_t||_2^2\mathbf{1}_{\{\langle\nabla f(\theta^t), s_t\rangle\leq 0\}}\mathbf{1}_{\{\nabla f(\theta^t)\neq 0\}} \mid \theta^t\big].$$

Since the PDF of the distribution $\mathcal{N}(0, I_d)$ is origin-symmetric, we have: $\mathbb{E}\big[||s_t||_2^2\mathbf{1}_{\{\langle\nabla f(\theta^t), s_t\rangle\leq 0\}}\mathbf{1}_{\{\nabla f(\theta^t)\neq 0\}} \mid \theta^t\big] \leq \frac{\mathbb{E}\big[||s_t||_2^2\big]}{2} = \frac{d}{2}$.

If $\nabla f(\theta^t) \neq 0$, we have:

$$\begin{aligned}
\mathbb{E}\left[|\langle\nabla f\left(\theta^t\right), s_t\rangle| \mid \theta^t\right] &= \left\|\nabla f\left(\theta^t\right)\right\|_2 \mathbb{E}\left[\left|\left\langle\frac{\nabla f\left(\theta^t\right)}{\left\|\nabla f\left(\theta^t\right)\right\|_2}, s_t\right\rangle\right| \mid \theta^t\right] \\
&= \left\|\nabla f\left(\theta^t\right)\right\|_2 \mathbb{E}_{s\sim\mathcal{N}(0,1)}[|s|] \\
&= \frac{2}{\sqrt{2\pi}}\left\|\nabla f\left(\theta^t\right)\right\|_2 \int_0^\infty se^{-\frac{s^2}{2}}ds \\
&= \sqrt{\frac{2}{\pi}}\left\|\nabla f\left(\theta^t\right)\right\|_2.
\end{aligned}$$

If $\nabla f(\theta^t) = 0$, both sides of the equality above are trivially zero. Therefore, in both cases, we have: $\mathbb{E}\left[|\langle\nabla f\left(\theta^t\right), s_t\rangle| \mid \theta^t\right] = \sqrt{\frac{2}{\pi}}\left\|\nabla f\left(\theta^t\right)\right\|_2$, and it follows that:

$$\frac{1}{\sqrt{2\pi}}\alpha_t\left\|\nabla f\left(\theta^t\right)\right\|_2 \leq \mathbb{E}\left[f\left(\theta^t\right) - f\left(\theta^{t+1}\right) \mid \theta^t\right] + \frac{Ld\alpha_t^2}{4}.$$

We conclude that:

$$\frac{1}{\sqrt{2\pi}}\alpha_t\mathbb{E}\left[\left\|\nabla f\left(\theta^t\right)\right\|_2\right] \leq \mathbb{E}\left[f\left(\theta^t\right)\right] - \mathbb{E}\left[f\left(\theta^{t+1}\right)\right] + \frac{Ld\alpha_t^2}{4}.$$

$\square$

*Proof of Theorem 2.* Using lemma 1, for all $t \geq 1$, we have:

$$\frac{1}{\sqrt{2\pi}}\alpha_t\mathbb{E}\left[\left\|\nabla f\left(\theta^t\right)\right\|_2\right] \leq \mathbb{E}\left[f\left(\theta^t\right)\right] - \mathbb{E}\left[f\left(\theta^{t+1}\right)\right] + \frac{Ld\alpha_t^2}{4}.$$

Define the function $g$ as follows: $\forall\theta \in \mathbb{R}^d, g(\theta) = f(\theta) - \inf_{\theta'\in\mathbb{R}^d} f(\theta')$. For all $t \geq 1$, we have:

$$\frac{\alpha_0}{\sqrt{2\pi d}}\mathbb{E}\left[\|\nabla f(\theta^t)\|_2\right] \leq \sqrt{t}\left(\mathbb{E}[g(\theta^t)] - \mathbb{E}[g(\theta^{t+1})]\right) + \frac{L\alpha_0^2}{4\sqrt{t}}.$$

For $T \geq 2$, summing over $t$, we obtain:

$$\frac{\alpha_0}{\sqrt{2\pi d}} \sum_{t=1}^{T} \mathbb{E}\left[\|\nabla f(\theta^t)\|_2\right] \leq \sum_{t=1}^{T} \sqrt{t}\,\mathbb{E}[g(\theta^t)] - \sum_{t=2}^{T+1} \sqrt{t-1}\,\mathbb{E}[g(\theta^t)] + \frac{L\alpha_0^2}{4} \sum_{t=1}^{T} \frac{1}{\sqrt{t}}$$

$$\leq g(\theta^1) - \sqrt{T}\,\mathbb{E}[g(\theta^{T+1})] + \sum_{t=2}^{T} \frac{\mathbb{E}[g(\theta^t)]}{\sqrt{t}+\sqrt{t-1}} + \frac{L\alpha_0^2}{2} \sum_{t=1}^{T} \int_{t-1}^{t} \frac{dx}{2\sqrt{x}}$$

$$\leq g(\theta^1) - \sqrt{T}\,\mathbb{E}[g(\theta^{T+1})] + g(\theta^1) \sum_{t=2}^{T} \frac{1}{2\sqrt{t-1}} + \frac{L\alpha_0^2}{2}\sqrt{T}$$

$$= g(\theta^1) - \sqrt{T}\,\mathbb{E}[g(\theta^{T+1})] + g(\theta^1) \sum_{t=1}^{T-1} \frac{1}{2\sqrt{t}} + \frac{L\alpha_0^2}{2}\sqrt{T}$$

$$\leq g(\theta^1) - \sqrt{T}\,\mathbb{E}[g(\theta^{T+1})] + g(\theta^1) \sum_{t=1}^{T-1} \int_{t-1}^{t} \frac{1}{2\sqrt{x}}\,dx + \frac{L\alpha_0^2}{2}\sqrt{T}$$

$$\leq g(\theta^1) - \sqrt{T}\,\mathbb{E}[g(\theta^{T+1})] + g(\theta^1)\sqrt{T-1} + \frac{L\alpha_0^2}{2}\sqrt{T}$$

$$\leq g(\theta^1) + \left(g(\theta^1) - \mathbb{E}[g(\theta^{T+1})]\right)\sqrt{T} + \frac{L\alpha_0^2}{2}\sqrt{T}$$

$$= f(\theta^1) - \inf_{\theta\in\mathbb{R}^d} f(\theta) + \left(f(\theta^1) - f(\theta^{T+1})\right)\sqrt{T} + \frac{L\alpha_0^2}{2}\sqrt{T}$$

$$\leq 2\left(f(\theta^1) - \inf_{\theta\in\mathbb{R}^d} f(\theta)\right)\sqrt{T} + \frac{L\alpha_0^2}{2}\sqrt{T}.$$

Thus, we conclude:

$$\frac{\sum_{t=1}^{T} \mathbb{E}\left[\|\nabla f(\theta^t)\|_2\right]}{T} \leq \left(\frac{2\sqrt{2\pi}\left(f(\theta^1) - \inf_{\theta\in\mathbb{R}^d} f(\theta)\right)}{\alpha_0} + \sqrt{\frac{\pi}{2}}L\alpha_0\right)\sqrt{\frac{d}{T}}.$$

$\square$

### A.2 Convergence analysis for pMSS algorithm

**Lemma 3.** *Assume that $f$ is lower bounded. Under algorithm 2, for all $k \geq 1$, we have*

$$\mathbb{P}\left(\tau_k < \infty \cap \rho_k = \infty\right) = 0.$$

*Equivalently, on the event $\{\tau_k < \infty\}$ we have $\rho_k < \infty$ almost surely.*

*Proof of Lemma 3.* Fix $k \geq 1$ and define the event $B_k := \{\tau_k < \infty\} \cap \{\rho_k = \infty\}$. We will show that $B_k$ is empty, leading to the desired result. Let $\omega \in B_k$. By definition of $B_k$ we have

$$\tau_k(\omega) < \infty \quad \text{and} \quad \rho_k(\omega) = \infty.$$

Then have, for all $t \geq \tau_k(\omega)$,

$$f(\theta^{t+1}(\omega)) \leq f(\theta^t(\omega)) - c\,a_k^2.$$

It follows that for all $n \geq 1$, we have:

$$f\left(\theta^{\tau_k(\omega)+n}(\omega)\right) \leq f\left(\theta^{\tau_k(\omega)}(\omega)\right) - n\,c\,a_k^2. \tag{2}$$

Since $f$ is bounded below on $\mathbb{R}^d$, letting $n \to \infty$ yields a contradiction.

Therefore no such $\omega$ can exist, and we conclude that

$$B_k = \varnothing, \quad \text{hence} \quad \mathbb{P}(B_k) = 0.$$

Equivalently, on the event $\{\tau_k < \infty\}$ we must have $\rho_k < \infty$ almost surely. $\square$

Using lemma 3 we can show that a new Gaussian direction is sampled infinitely many times almost surely.

**Lemma 4.** *Assume that $f$ is lower bounded. Under algorithm 2, we have almost surely:*

$$(i) \ \tau_k < \infty \ \text{for all } k, \qquad (ii) \ \tau_k \to \infty \ \text{as } k \to \infty.$$

*In particular, resampling occurs infinitely many times almost surely.*

*Proof of Lemma 4.* Using Lemma 3, we have:

$$\mathbb{P}\big(\tau_k < \infty \cap \rho_k = \infty \big) = 0 \quad \text{for all } k \geq 1.$$

**Proof of (i).** For each $k \geq 1$, define the event

$$E_k := \big\{\tau_k < \infty \ \Rightarrow \ \rho_k < \infty\big\}.$$

Since $\mathbb{P}\big(\rho_k = \infty \cap \tau_k < \infty\big) = 0$, we have $\mathbb{P}(E_k) = 1$ for every $k \geq 1$. Let $E := \bigcap_{k=1}^{\infty} E_k$. We have $\mathbb{P}(E) = 1$ because $E$ is a countable intersection of events of probability one.

Fix $\omega \in E$. We now argue pathwise.

*Base case.* By definition, $\tau_1(\omega) = 1 < \infty$.

*Induction step.* Suppose $\tau_k(\omega) < \infty$ for some $k \geq 1$. Since $\omega \in E_k$, the implication

$$\tau_k(\omega) < \infty \ \Rightarrow \ \rho_k(\omega) < \infty$$

holds, hence $\rho_k(\omega) < \infty$. By the definition of $\tau_{k+1}$,

$$\tau_{k+1}(\omega) = \tau_k + \rho_k(\omega) + 1 < \infty.$$

Thus, by induction, $\tau_k(\omega) < \infty$ for all $k \geq 1$.

Since this holds for every $\omega \in E$ and $\mathbb{P}(E) = 1$, we conclude that $\mathbb{P}\big(\tau_k < \infty \ \text{for all } k\big) = 1$, which proves (i).

**Proof of (ii).** Again we work on the event $E$ of probability one, on which all $\tau_k(\omega)$ are finite.

Fix $\omega \in E$. For each $k \geq 1$ we have,

$$\tau_{k+1}(\omega) = \tau_k(\omega) + \rho_k(\omega) + 1 \ \geq \ \tau_k(\omega) + 1.$$

Hence the sequence $\{\tau_k(\omega)\}_{k \geq 1}$ is an increasing sequence of integers, so

$$\tau_k(\omega) \xrightarrow[k \to \infty]{} \infty.$$

Since this holds for every $\omega \in E$ and $\mathbb{P}(E) = 1$, we obtain

$$\tau_k \to \infty \quad \text{as } k \to \infty \quad \text{almost surely,}$$

which proves (ii).

Finally, note that at each time $\tau_k$ the algorithm draws a fresh Gaussian direction $s_{\tau_k} \sim \mathcal{N}(0, I_d)$. Since $\tau_k < \infty$ for all $k$ and $\tau_k \to \infty$ almost surely, it follows that resampling occurs infinitely many times almost surely. $\qquad \square$

*Proof of Theorem 3.* Using Lemma 4, almost surely, for all $k \geq 1$ we have $\tau_k < \infty$. Let $t \geq 1$. Using the monotonic improvement of the algorithm and assuming that $f$ is $L$-smooth, we have almost surely

$$f(\theta^{\tau_{t+1}}) \leq f(\theta^{\tau_t + 1}) \leq f(\theta^{\tau_t} + \beta_{\tau_t} s_{\tau_t}) \leq f(\theta^{\tau_t}) + \beta_{\tau_t} \langle \nabla f(\theta^{\tau_t}), s_{\tau_t} \rangle + \frac{L}{2} \beta_{\tau_t}^2 \|s_{\tau_t}\|_2^2.$$

Since $s_{\tau_t}$ is a fresh Gaussian, independent of the filtration $\mathcal{F}_{\tau_t}$, we can repeat the proof of Lemma 1 to get

$$\frac{1}{\sqrt{2\pi}} a_t \mathbb{E}\left[\|\nabla f(\theta^{\tau_t})\|_2\right] \leq \mathbb{E}\left[f(\theta^{\tau_t})\right] - \mathbb{E}\left[f(\theta^{\tau_{t+1}})\right] + \frac{Lda_t^2}{4}. \tag{3}$$

Summing from $t = 1$ to $N$ yields

$$\frac{1}{\sqrt{2\pi}} \sum_{t=1}^{N} a_t \, \mathbb{E}\Big[\big\|\nabla f\big(\theta^{\tau_t}\big)\big\|_2\Big] \ \leq \ \mathbb{E}\big[f(\theta^{\tau_1})\big] - \mathbb{E}\big[f(\theta^{\tau_{N+1}})\big] + \frac{Ld}{4} \sum_{t=1}^{N} a_t^2.$$

By monotonicity of the algorithm, $f(\theta^t)$ is nonincreasing and hence $\mathbb{E}\big[f(\theta^{\tau_1})\big] \leq f(\theta^1)$. If $f$ is bounded below by $f_\star > -\infty$ then $\mathbb{E}[f(\theta^{\tau_{N+1}})] \geq f_\star$ for all $N$, and therefore

$$\frac{1}{\sqrt{2\pi}} \sum_{t=1}^{N} a_t \, \mathbb{E}\Big[\big\|\nabla f\big(\theta^{\tau_t}\big)\big\|_2\Big] \ \leq \ f(\theta^1) - f_\star + \frac{Ld}{4} \sum_{t=1}^{\infty} a_t^2.$$

Letting $N \to \infty$ and using $\sum_{t=1}^{\infty} a_t^2 < \infty$, we obtain

$$\sum_{t=1}^{\infty} a_t \, \mathbb{E}\Big[\big\|\nabla f\big(\theta^{\tau_t}\big)\big\|_2\Big] \ < \ \infty. \tag{4}$$

Define the nonnegative random variable $S(\omega) := \sum_{t=1}^{\infty} a_t \, \big\|\nabla f\big(\theta^{\tau_t}(\omega)\big)\big\|_2 \in [0, +\infty]$.

By equation 4, we have: $\mathbb{E}[S] = \sum_{t=1}^{\infty} a_t \, \mathbb{E}\Big[\big\|\nabla f\big(\theta^{\tau_t}\big)\big\|_2\Big] < \infty$. Since $S \geq 0$ and $\mathbb{E}[S] < \infty$, we must have $\mathbb{P}(S = +\infty) = 0$.

Fix $\varepsilon > 0$ and define

$$B_\varepsilon := \Big\{ \exists K \geq 1 \text{ such that } \big\|\nabla f\big(\theta^{\tau_t}\big)\big\|_2 \geq \varepsilon \text{ for all } t \geq K \Big\}.$$

Suppose, for a contradiction, that $\mathbb{P}(B_\varepsilon) > 0$. For every $\omega \in B_\varepsilon$ there exists $K(\omega) \geq 1$ such that $\|\nabla f(\theta^{\tau_t}(\omega))\|_2 \geq \varepsilon$ for all $t \geq K(\omega)$, hence

$$S(\omega) = \sum_{t=1}^{\infty} a_t \, \big\|\nabla f\big(\theta^{\tau_t}(\omega)\big)\big\|_2 \ \geq \ \sum_{t=K(\omega)}^{\infty} a_t \, \varepsilon = \varepsilon \sum_{t=K(\omega)}^{\infty} a_t = +\infty,$$

because $\sum_{t=1}^{\infty} a_t = \infty$ by assumption. Thus $S(\omega) = +\infty$ for all $\omega \in B_\varepsilon$, which implies $\mathbb{P}(S = +\infty) \geq \mathbb{P}(B_\varepsilon) > 0$, contradicting $\mathbb{P}(S = +\infty) = 0$. Therefore $\mathbb{P}(B_\varepsilon) = 0$ for every $\varepsilon > 0$, and hence:

$$\liminf_{t \to \infty} \big\|\nabla f\big(\theta^{\tau_t}\big)\big\|_2 = 0 \quad \text{almost surely.}$$

To extend this from the resampling times $\{\tau_t\}$ to all iterations, define

$$C_\varepsilon := \Big\{ \exists T \geq 1 \text{ such that } \|\nabla f(\theta^t)\|_2 \geq \varepsilon \text{ for all } t \geq T \Big\}.$$

Assume by contradiction that $\mathbb{P}(C_\varepsilon) > 0$ for some $\varepsilon > 0$. By denoting $D_\varepsilon = C_\varepsilon \cap \{\tau_t \to \infty\}$, using Lemma 4, we have $\mathbb{P}(D_\varepsilon) = \mathbb{P}(C_\varepsilon) > 0$. Let $\omega \in D_\varepsilon$. There exist $T(\omega), K(\omega)$ such that $\|\nabla f(\theta^t(\omega))\|_2 \geq \varepsilon$ for all $t \geq T(\omega)$ and $\tau_t(\omega) \geq T(\omega)$ for all $t \geq K(\omega)$. Thus

$$\big\|\nabla f\big(\theta^{\tau_t}(\omega)\big)\big\|_2 \ \geq \ \varepsilon \quad \text{for all } t \geq K(\omega).$$

This means $D_\varepsilon \subseteq B_\varepsilon$, and therefore $\mathbb{P}(D_\varepsilon) \leq \mathbb{P}(B_\varepsilon) = 0$, a contradiction. Hence $\mathbb{P}(C_\varepsilon) = 0$ for every $\varepsilon > 0$, which is equivalent to

$$\liminf_{t \to \infty} \|\nabla f(\theta^t)\|_2 = 0 \quad \text{almost surely.}$$

This proves the theorem. $\qquad\qquad\qquad\qquad\qquad\qquad\qquad\qquad\qquad\qquad\qquad\qquad\square$

## A.3 Convergence analysis for MSS algorithm with stochastic ranking oracle

**Lemma 5.** *Assume that $f$ is L-smooth. Under Algorithm 3, for all $t \geq 1$,*

$$\mathbb{E}\big[\Delta_t \, \mathbf{1}_{B_t} \,\big|\, \theta^t\big] \ \leq \ -\frac{\alpha_t}{\sqrt{2\pi}} \, \|\nabla f(\theta^t)\|_2 \ + \ \frac{L}{2} \, d \, \alpha_t^2.$$

*Proof of Lemma 5.* By $L$-smoothness, we have: $\Delta_t \leq \alpha_t \langle \nabla f(\theta^t), s_t \rangle + \frac{L}{2}\alpha_t^2 \|s_t\|^2$. Therefore

$$\Delta_t \, \mathbf{1}_{B_t} \leq \min(\Delta_t, 0) \leq \min(\alpha_t \langle \nabla f(\theta^t), s_t \rangle, 0) + \frac{L}{2}\alpha_t^2 \|s_t\|_2^2,$$

where we used $\min(y + b, 0) \leq \min(y, 0) + b$ for any $b \geq 0$. Taking conditional expectation and using $s_t \sim \mathcal{N}(0, I_d)$,

$$\mathbb{E}\big[\Delta_t \, \mathbf{1}_{B_t} \,\big|\, \theta^t\big] \leq \alpha_t \, \mathbb{E}\big[\min\{\langle \nabla f(\theta^t), s_t \rangle, 0\} \,\big|\, \theta^t\big] + \frac{L}{2}\alpha_t^2 \, \mathbb{E}\big[\|s_t\|_2^2\big]$$

$$= \alpha_t \, \mathbb{E}\left[ \frac{\langle \nabla f(\theta^t), s_t \rangle - \big|\langle \nabla f(\theta^t), s_t \rangle\big|}{2} \,\middle|\, \theta^t \right] + \frac{L}{2}\, d\, \alpha_t^2$$

$$= -\frac{\alpha_t}{2} \, \mathbb{E}\big[\big|\langle \nabla f(\theta^t), s_t \rangle\big| \,\big|\, \theta^t\big] + \frac{L}{2}\, d\, \alpha_t^2$$

$$= \frac{-\alpha_t}{\sqrt{2\pi}} \, \big\|\nabla f\big(\theta^t\big)\big\|_2 + \frac{L}{2}\, d\, \alpha_t^2.$$

$\square$

The next lemma shows that the probability of a wrong accept/reject decision decays exponentially in both the number of votes $N$ and the margin $h(|\Delta_t|)$.

**Lemma 6.** *Assume assumption 1. Conditionally on $(\theta^t, s_t)$ and when $\Delta_t \neq 0$,*

$$\mathbb{E}\big[\big|\, \mathbf{1}_{A_t} - \mathbf{1}_{B_t}\,\big| \,\big|\, \theta^t, s_t\big] \leq \exp\!\Big(-2N\, h(|\Delta_t|)^2\Big).$$

*Proof of lemma 6.* By definition, we have: $\big|\mathbf{1}_{A_t} - \mathbf{1}_{B_t}\big| = \mathbf{1}_{\{A_t \triangle B_t\}}$, so that:

$$\mathbb{E}\big[\big|\mathbf{1}_{A_t} - \mathbf{1}_{B_t}\big| \,\big|\, \theta^t, s_t\big] = \mathbb{P}\big(A_t \triangle B_t \,\big|\, \theta^t, s_t\big).$$

Write $\Delta_t = f(\theta^t + \alpha_t s_t) - f(\theta^t)$ and $p_t := \mathbb{P}(o_{t,1} = 1 \mid \theta^t, s_t)$. By independence the $o_{t,1}, \ldots, o_{t,N}$ are i.i.d. Bernoulli($p_t$) conditional on $\theta^t, s_t$.

Assume now that $\Delta_t < 0$. By the oracle assumption: $\mathbb{P}(o_{t,1} = 1 \mid \theta^t, s_t) \geq \frac{1}{2} + h(|\Delta_t|)$. It holds that:

$$\mathbb{P}\big(\bar{o}_t - \tfrac{1}{2} \leq 0 \,\big|\, \theta^t, s_t\big) = \mathbb{P}\left( \frac{1}{N}\sum_{n=1}^{N}\Big(\mathbf{1}_{\{o_{t,n}=1\}} - \tfrac{1}{2}\Big) \leq 0 \,\middle|\, \theta^t, s_t \right)$$

$$\leq \mathbb{P}\left( \frac{1}{N}\sum_{n=1}^{N}\Big(\mathbf{1}_{\{o_{t,n}=1\}} - \tfrac{1}{2}\Big) \leq \underbrace{\mathbb{E}\big[\mathbf{1}_{\{o_{t,1}=1\}} \mid \theta^t, s_t\big] - \tfrac{1}{2} - h(\Delta_t)}_{\geq 0} \,\middle|\, \theta^t, s_t \right)$$

$$= \mathbb{P}\left( \frac{1}{N}\sum_{n=1}^{N}\Big(\mathbf{1}_{\{o_{t,n}=1\}} - \tfrac{1}{2}\Big) - \Big(\mathbb{E}\big[\mathbf{1}_{\{o_{t,1}=1\}} \mid \theta^t, s_t\big] - \tfrac{1}{2}\Big) \leq -h(\Delta_t) \,\middle|\, \theta^t, s_t \right)$$

$$\leq \exp\big(-2N\, h(|\Delta_t|)^2\big) \qquad \text{(by Hoeffding's inequality).}$$

But $\{\bar{o}_t \leq \tfrac{1}{2}\}$ is exactly the misclassification event $A_t \triangle B_t$ in this case.

Assume now that $\Delta_t > 0$. By the oracle assumption: $\mathbb{P}(o_{t,1} = 0 \mid \theta^t, s_t) \geq \frac{1}{2} + h(|\Delta_t|)$. It holds that:

$$\mathbb{P}\big(\bar{o}_t - \tfrac{1}{2} > 0 \,\big|\, \theta^t, s_t\big) = \mathbb{P}\left( \frac{1}{N}\sum_{n=1}^{N}\Big(\tfrac{1}{2} - \mathbf{1}_{\{o_{t,n}=1\}}\Big) < 0 \,\middle|\, \theta^t, s_t \right)$$

$$\leq \mathbb{P}\left( \frac{1}{N}\sum_{n=1}^{N}\Big(\tfrac{1}{2} - \mathbf{1}_{\{o_{t,n}=1\}}\Big) \leq \underbrace{\mathbb{E}\big[\mathbf{1}_{\{o_{t,n}=0\}} \mid \theta^t, s_t\big] - \tfrac{1}{2} - h(\Delta_t)}_{\geq 0} \,\middle|\, \theta^t, s_t \right)$$

$$= \mathbb{P}\left( \frac{1}{N}\sum_{n=1}^{N}\Big(\tfrac{1}{2} - \mathbf{1}_{\{o_{t,n}=1\}}\Big) - \Big(\tfrac{1}{2} - \mathbb{E}\big[\mathbf{1}_{\{o_{t,n}=1\}} \mid \theta^t, s_t\big]\Big) \leq -h(\Delta_t) \,\middle|\, \theta^t, s_t \right)$$

$$\leq \exp\big(-2N\, h(\Delta_t)^2\big) \qquad \text{(by Hoeffding's inequality).}$$

which is the misclassification event $A_t \triangle B_t$ in this case. Combining both cases we obtain the desired result. $\square$

The next lemma upper-bounds the ranking-error term.

---

**Lemma 7.** *Assume that $f$ is $L-$smooth and assumption 1 holds. Under algorithm 3, for all $t \geq 1$, we have:*

$$\left| \mathbb{E}[\Delta_t(\mathbf{1}_{A_t} - \mathbf{1}_{B_t}) \mid \theta^t] \right| \leq \frac{C_{p,\kappa}}{N^{\frac{1}{2p}}} + e^{-2Nm_r^2}\left( \alpha_t \sqrt{\frac{2}{\pi}} \|\nabla f(\theta^t)\|_2 + \frac{L}{2} d\, \alpha_t^2 \right),$$

*where $C_{p,\kappa} := e^{-\frac{1}{2p}} \left( 4p\, \kappa^2 \right)^{-\frac{1}{2p}}$.*

---

*Proof of Lemma 7.* Condition on $(\theta^t, s_t)$. By Lemma 6 and Assumption 1, if $\Delta_t \neq 0$, we have:

$$\mathbb{E}\left[ \left| \mathbf{1}_{A_t} - \mathbf{1}_{B_t} \right| \mid \theta^t, s_t \right] \leq \exp\left( -2N\, h(|\Delta_t|)^2 \right) \leq e^{-2N\kappa^2|\Delta_t|^{2p}} \mathbf{1}_{\{|\Delta_t| \leq r\}} + e^{-2Nm_r^2} \mathbf{1}_{\{|\Delta_t| > r\}}.$$

then, if $\Delta_t \neq 0$, we have:

$$\mathbb{E}\left[ |\Delta_t| \left| \mathbf{1}_{A_t} - \mathbf{1}_{B_t} \right| \mid \theta^t, s_t \right] \leq \left| \Delta_t \right| e^{-2N\kappa^2|\Delta_t|^{2p}} \mathbf{1}_{\{|\Delta_t| \leq r\}} + \left| \Delta_t \right| e^{-2Nm_r^2}.$$

We remark that the inequality above holds trivially if $\Delta_t = 0$. By taking expectation given $\theta^t$, we get:

$$\mathbb{E}\left[ |\Delta_t| \left| \mathbf{1}_{A_t} - \mathbf{1}_{B_t} \right| \mid \theta^t \right] \leq \sup_{x \in [0,r]} x\, e^{-2N\kappa^2 x^{2p}} + e^{-2Nm_r^2} \mathbb{E}\left[ |\Delta_t| \mid \theta^t \right].$$

The map $\phi(x) = x e^{-ax^{2p}}$ attains its maximum at $x^\star = (2pa)^{-1/(2p)}$ with value $e^{-1/(2p)}(2pa)^{-1/(2p)}$. With $a = 2N\kappa^2$ we obtain the first term as $C_{p,\kappa} N^{-1/(2p)}$. For the second term, $L$-smoothness gives $|\Delta_t| \leq \alpha_t |\langle \nabla f(\theta^t), s_t \rangle| + \frac{L}{2}\alpha_t^2 \|s_t\|_2^2$, and since $s_t \sim \mathcal{N}(0, I_d)$, $\mathbb{E}[|\langle \nabla f(\theta^t), s_t \rangle| \mid \theta^t] = \sqrt{\frac{2}{\pi}} \|\nabla f(\theta^t)\|_2$ and $\mathbb{E}[\|s_t\|_2^2] = d$. Combine the bounds. $\square$

Putting lemma 5 and lemma 7 together gives a descent inequality for Algorithm 3.

---

**Lemma 8.** *Assume that $f$ is $L-$smooth and assumption 1 holds. Under algorithm 3, for all $t \geq 1$, we have:*

$$\frac{\alpha_t}{\sqrt{2\pi}}\left( 1 - 2e^{-2Nm_r^2} \right) \mathbb{E}\left[ \|\nabla f(\theta^t)\|_2 \right] \leq \mathbb{E}\left[ f(\theta^t) - f(\theta^{t+1}) \right] + \frac{L}{2} d\, \alpha_t^2 \left( 1 + e^{-2Nm_r^2} \right) + \frac{C_{p,\kappa}}{N^{\frac{1}{2p}}}.$$

---

*Proof of Theorem 4.* It is a direct consequence of Lemma 8. $\square$

### A.3.1 CONVERGENCE ANALYSIS FOR MSS ALGORITHM WITH GRADIENT APPROXIMATION

*Proof of Lemma 2.* For all $\theta \in \mathbb{R}^d$, we denote:

$$\begin{cases} \tilde{\nabla}_{s_t} f(\theta) := \frac{f(\theta^t + \gamma_t s_t) - f(\theta^t)}{\gamma_t} \\ \nabla_{s_t} f(\theta) := \langle \nabla f(\theta^t), s_t \rangle \end{cases}.$$

Let $t \geq 1$. Using the smoothness of $f$ and the monotonic improvement property of the algorithm, we obtain:

$$\mathbb{E}[f(\theta^{t+1}) \mid \theta^t] \leq \mathbb{E}\left[f\left(\theta^t - \alpha \frac{f(\theta^t + \gamma_t s_t) - f(\theta^t)}{\gamma_t} s_t\right) \mid \theta^t\right]$$

$$= \mathbb{E}\left[f\left(\theta^t - \alpha \tilde{\nabla}_{s_t} f(\theta^t) s_t\right) \mid \theta^t\right]$$

$$\leq \mathbb{E}\left[f(\theta^t) - \alpha \tilde{\nabla}_{s_t} f(\theta^t)\langle \nabla f(\theta^t), s_t \rangle + \frac{L}{2}\left(\alpha \tilde{\nabla}_{s_t} f(\theta^t)\|s_t\|_2\right)^2 \mid \theta^t\right] \text{ (smoothness)}$$

$$= \mathbb{E}\left[f(\theta^t) - \alpha \tilde{\nabla}_{s_t} f(\theta^t) \nabla_{s_t} f(\theta^t) + \frac{L}{2}\left(\alpha \tilde{\nabla}_{s_t} f(\theta^t)\right)^2 \mid \theta^t\right]$$

$$= \mathbb{E}\left[f(\theta^t) + \alpha\left(\frac{(\tilde{\nabla}_{s_t} f(\theta^t) - \nabla_{s_t} f(\theta^t))^2}{2} - \frac{(\tilde{\nabla}_{s_t} f(\theta^t))^2 + (\nabla_{s_t} f(\theta^t))^2}{2}\right)\right.$$

$$\left. + \frac{L}{2}\left(\alpha \tilde{\nabla}_{s_t} f(\theta^t)\right)^2 \mid \theta^t\right]$$

$$= \mathbb{E}\left[f(\theta^t) - \frac{\alpha}{2}(\nabla_{s_t} f(\theta^t))^2 + \frac{\alpha}{2}(\tilde{\nabla}_{s_t} f(\theta^t) - \nabla_{s_t} f(\theta^t))^2\right.$$

$$\left. + (\tilde{\nabla}_{s_t} f(\theta^t))^2 \left(\frac{\alpha(L\alpha - 1)}{2}\right) \mid \theta^t\right]$$

$$\leq \mathbb{E}\left[f(\theta^t) - \frac{\alpha}{2}(\nabla_{s_t} f(\theta^t))^2 + \frac{\alpha}{2}(\tilde{\nabla}_{s_t} f(\theta^t) - \nabla_{s_t} f(\theta^t))^2 \mid \theta^t\right].$$

Using the smoothness of $f$, we obtain the following bound:

$$\left|f(\theta^t + \gamma_t s_t) - f(\theta^t) - \langle \gamma_t \nabla f(\theta^t), s_t \rangle\right| \leq \frac{L}{2}\gamma_t^2 \|s_t\|_2^2.$$

This implies:

$$\mathbb{E}\left[\left|\tilde{\nabla}_{s_t} f(\theta^t) - \nabla_{s_t} f(\theta^t)\right|^2 \Big| \theta^t\right] \leq \frac{L^2}{4}\gamma_t^2. \tag{5}$$

Therefore:

$$\frac{\alpha}{2}\mathbb{E}[\langle \nabla f(\theta^t), s_t \rangle^2 \mid \theta^t] = \frac{\alpha}{2}\mathbb{E}[(\nabla_{s_t} f(\theta^t))^2 \mid \theta^t] \leq \mathbb{E}[f(\theta^t) - f(\theta^{t+1}) \mid \theta^t] + \frac{\alpha L^2}{8}\gamma_t^2. \tag{6}$$

Assume that $\theta^t \in \{v \in \mathbb{R}^d \mid \nabla f(v) \neq 0\}$. We have:

$$\mathbb{E}\left[\left|\langle \nabla f(\theta^t), s_t \rangle\right|^2 \Big| \theta^t\right] = \|\nabla f(\theta^t)\|_2^2 \, \mathbb{E}\left[\left|\left\langle \frac{\nabla f(\theta^t)}{\|\nabla f(\theta^t)\|_2}, s_t \right\rangle\right|^2 \Big| \theta^t\right].$$

Let $R(\theta^t)$ be an orthogonal matrix such that $R(\theta^t)\frac{\nabla f(\theta^t)}{\|\nabla f(\theta^t)\|_2} = e_1$. We have:

$$\mathbb{E}\left[\left|\langle \nabla f(\theta^t), s_t \rangle\right|^2 \Big| \theta^t\right] = \|\nabla f(\theta^t)\|_2^2 \, \mathbb{E}\left[\left|\langle R(\theta^t)^\top e_1, s_t \rangle\right|^2 \Big| \theta^t\right]$$

$$= \|\nabla f(\theta^t)\|_2^2 \, \mathbb{E}\left[\left|\langle e_1, R(\theta^t) s_t \rangle\right|^2 \Big| \theta^t\right]$$

$$= \|\nabla f(\theta^t)\|_2^2 \, \mathbb{E}\left[\left|\langle e_1, s_t \rangle\right|^2\right]$$

$$= \frac{1}{d}\|\nabla f(\theta^t)\|_2^2 \, \mathbb{E}[\|s_t\|_2^2]$$

$$= \frac{1}{d}\|\nabla f(\theta^t)\|_2^2.$$

This implies that $\mathbb{E}\left[\langle \nabla f(\theta^t), s_t \rangle^2 \mid \theta^t\right] = \frac{1}{d}\|\nabla f(\theta^t)\|_2^2$. Assuming $\theta^t \notin \{\theta \in \mathbb{R}^d \mid \nabla f(\theta) \neq 0\}$, the inequality still holds. Combining this with inequality (6), we obtain:

$$\frac{\alpha}{2d}\|\nabla f(\theta^t)\|_2^2 \leq \mathbb{E}[f(\theta^t) - f(\theta^{t+1}) \mid \theta^t] + \frac{\alpha L^2}{8}\gamma_t^2.$$

By taking expectation, we get:

$$\mathbb{E}[\|\nabla f(\theta^t)\|_2^2] \leq \frac{2d(\mathbb{E}[f(\theta^t)] - \mathbb{E}[f(\theta^{t+1})])}{\alpha} + \frac{dL^2}{4}\gamma_t^2.$$

$\square$

*Proof of Theorem 6.* Let $X_T = \left( \min_{1 \leq t \leq T} \|\nabla f(\theta^t)\|_2 \right)^2$ for all $T \geq 1$. Since $\sum_{t=1}^{\infty} \gamma_t^2 < \infty$, using lemma 2, we have $\mathbb{E}[\sum_{t=1}^{\infty} X_t] = \sum_{t=1}^{\infty} \mathbb{E}[X_t] < \infty$. Then $\sum_{t=1}^{\infty} X_t < \infty$ almost surely.

Now fix $T \geq 1$. Since $\{X_t\}_{t \geq 1}$ is non-increasing, we have:

$$TX_{2T} \leq \sum_{i=T}^{2T-1} X_i \leq \sum_{i=T}^{\infty} X_i.$$

Since $\lim_{T \to \infty} \sum_{i=T}^{\infty} X_i = 0$ almost surely, it holds that:

$$TX_{2T} \to 0 \quad \text{almost surely, i.e.,} \quad (2T)X_{2T} = o(1) \quad \text{a.s.} .$$

A similar argument gives $(2T+1)X_{2T+1} = o(1)$ almost surely.

Combining these results, we deduce that:

$$X_T = o\left( \frac{1}{T} \right) \quad \text{almost surely.}$$

$\square$

# B  CONVERGENCE ANALYSIS FOR MSS ALGORITHM IN THE NON-SMOOTH SETTING

In this appendix we provide the auxiliary lemmas and proof of theorem 7. Throughout we assume assumption 2 and that $d \geq 3$.

Recall the set of good directions

$$A_d(\theta) := \left\{ s \in \mathbb{S}^{d-1} : \langle \nabla f(\theta), s \rangle \leq -\frac{1}{2\sqrt{d}} \|\nabla f(\theta)\|_2 \right\},$$

and let $\mathcal{U}(\mathbb{S}^{d-1})$ denote the uniform distribution on the unit sphere.

---

**Lemma 9.** *Let $d \geq 3$. For every $\theta \in \mathbb{R}^d$, we have*

$$p_d := \mathbb{P}_{s \sim \mathcal{U}(\mathbb{S}^{d-1})}\{ s \in A_d(\theta) \} \geq \frac{1}{4}.$$

---

*Proof of Lemma 9.* Fix any unit vector $u$ and let $Z = \langle u, s \rangle$ with $s \sim \mathcal{U}(\mathbb{S}^{d-1})$. Then, using (Vignat & Plastino, 2005, Theorem 2), $Z$ has density on $(-1, 1)$:

$$f_d(z) = c_d (1 - z^2)^{\frac{d-3}{2}}, \qquad c_d = \frac{\Gamma(\frac{d}{2})}{\sqrt{\pi}\,\Gamma(\frac{d-1}{2})}.$$

For $d \geq 3$, $f_d$ is even and nonincreasing on $[0, 1)$. Then:

$$p_d = \mathbb{P}\{Z \leq -\frac{1}{2\sqrt{d}}\} = \frac{1}{2} - \int_0^{\frac{1}{2\sqrt{d}}} f_d(z)\,dz \geq \frac{1}{2} - \frac{1}{2\sqrt{d}} f_d(0) = \frac{1}{2} - \frac{1}{2\sqrt{d}} c_d.$$

Using Wendel's inequality, $\frac{\Gamma(\frac{d}{2})}{\Gamma(\frac{d-1}{2})} \leq \sqrt{\frac{d-1}{2}} \leq \sqrt{\frac{d}{2}}$, we have $c_d \leq \sqrt{\frac{d}{2\pi}}$. Hence

$$\frac{1}{2\sqrt{d}} c_d \leq \frac{1}{2\sqrt{d}} \sqrt{\frac{d}{2\pi}} = \frac{1}{2\sqrt{2\pi}} \leq \frac{1}{4},$$

We conclude that $p_d \geq \frac{1}{4}$. $\square$

*Proof.* $\square$

**Lemma 10.** *Assume that assumption 2 holds. For all $\epsilon > 0$, there exists $r_\epsilon > 0$ such that, for all $\theta \in \mathcal{L}(\theta^1)$, all $s \in A_d(\theta)$, and all $\alpha \in (0, r_\epsilon)$, we have*

$$\|\nabla f(\theta)\|_2 \geq \epsilon \implies f(\theta + \alpha s) \ \leq \ f(\theta) \ - \ \frac{1}{4\sqrt{d}} \alpha \, \|\nabla f(\theta)\|_2.$$

*Proof of Lemma 10.* Let $K := \{x : \inf_{y \in \mathcal{L}(\theta^1)} \|x - y\|_2 \leq 1\}$. $K$ is compact and $\nabla f$ is uniformly continuous on $K$. For $\alpha \in [0, 1]$ and any $\theta \in \mathcal{L}(\theta^1)$, $s \in \mathbb{S}^{d-1}$, $t \in [0, 1]$, the points $\theta$ and $\theta + t\alpha s \in K$. Define on $[0, 1]$, the function $\mu$ as follows:

$$\mu(\alpha) \ := \ \sup_{\substack{\theta \in \mathcal{L}(\theta^1), \ s \in \mathbb{S}^{d-1}, \\ t \in [0,1]}} \big\|\nabla f(\theta + t\alpha s) - \nabla f(\theta)\big\|_2,$$

then $\mu(\alpha) \to 0$ as $\alpha \downarrow 0$. In fact, since $\nabla f$ is continuous on the compact $K$, by Heine theorem, it is uniformly continuous, meaning that for any fixed $\delta > 0$, there exists $\eta > 0$, for all $x, y \in K$ such that $\|x - y\|_2 \leq$, we have $\|\nabla f(x) - \nabla f(y)\|_2 \leq \delta$. Let $0 \leq \alpha \leq \min(1, )$, we get that for all $\theta \in \mathcal{L}(\theta^1)$, $s \in \mathbb{S}^{d-1}$, and $t \in [0, 1]$, $\|\nabla f(\theta + t\alpha s) - \nabla f(\theta)\|_2 \leq \delta$. This implies that if $0 \leq \alpha \leq \min(1, )$, we have $\mu(\alpha) \leq \delta$. Which means that $\mu(\alpha) \to 0$ as $\alpha \downarrow 0$.

Fix $\theta \in \mathcal{L}(\theta^1)$ and $s \in A_d(\theta)$.

For any $\alpha > 0$,

$$f(\theta + \alpha s) - f(\theta) = \alpha \langle \nabla f(\theta), s \rangle + \alpha \int_0^1 \langle \nabla f(\theta + t\alpha s) - \nabla f(\theta), \ s \rangle \, dt$$

$$\leq \ -\frac{\alpha}{2\sqrt{d}} \, \|\nabla f(\theta)\|_2 + \alpha \mu(\alpha).$$

Let $\epsilon > 0$. Since $\mu(\alpha) \to 0$ as $\alpha \downarrow 0$, there exists $r_\epsilon > 0$ such that for all $\alpha \in (0, r_\epsilon)$, we have $\mu(\alpha) \leq \frac{\epsilon}{4\sqrt{d}}$. Then for all $\theta \in \mathcal{L}(\theta^1)$, all $s \in A_d(\theta)$, and all $\alpha \in (0, r_\epsilon)$, if $\|\nabla f(\theta)\|_2 \geq \epsilon$ then we have:

$$f(\theta + \alpha s) \ \leq \ f(\theta) \ - \ \frac{\alpha}{4\sqrt{d}} \ \|\nabla f(\theta)\|_2.$$

$\square$

**Lemma 11.** *Assume that assumption 2 holds. Let $\{\theta^t\}$ be a sequence generated by algorithm 1 using the uniform distribution over the unit sphere. Let $\epsilon > 0$ and let $r_\epsilon > 0$ be as in Lemma 10. Then, for all $t$ such that $\alpha_t \leq r_\epsilon$,*

$$\mathbb{E}\big[f(\theta^{t+1}) \mid \theta^t\big] \ \leq \ f(\theta^t) - \frac{\alpha_t}{16\sqrt{d}} \|\nabla f(\theta^t)\|_2 \, \mathbf{1}_{\{\|\nabla f(\theta^t)\|_2 \geq \epsilon\}}.$$

*Proof of Lemma 11.* Let $E_t := \{\|\nabla f(\theta^t)\|_2 \geq \epsilon\}$ and $C_t := \{s_t \in A_d(\theta^t)\}$. If $\alpha_t \leq r_\epsilon$, Lemma 10 gives

$$f(\theta^{t+1}) \leq f(\theta^t + \alpha_t s_t) \ \leq \ f(\theta^t) \ - \ \frac{\alpha_t}{4\sqrt{d}} \|\nabla f(\theta^t)\| \quad \text{on } E_t \cap C_t.$$

Thus $f(\theta^{t+1}) \leq f(\theta^t) \ - \ \frac{\alpha_t}{4\sqrt{d}} \|\nabla f(\theta^t)\| \, \mathbf{1}_{E_t \cap C_t}$. We deduce that:

$$\mathbb{E}[f(\theta^{t+1}) \mid \theta^t] \leq f(\theta^t) - \frac{p_d \, \alpha_t}{4\sqrt{d}} \|\nabla f(\theta^t)\| \, \mathbf{1}_{E_t}$$

$$\leq f(\theta^t) - \frac{\alpha_t}{16\sqrt{d}} \|\nabla f(\theta^t)\| \, \mathbf{1}_{E_t}.$$

$\square$

We now prove the main convergence result.

*Proof of Theorem 7.* Fix $\epsilon > 0$. Since $\alpha_t \to 0$, there exists $T_\epsilon$ with $\alpha_t \leq r_\epsilon$ for all $t \geq T_\epsilon$. Taking expectations in Lemma 11 and summing from $t = T_\epsilon$ to $N - 1$, for any $N \geq T_\epsilon + 1$, yields

$$\mathbb{E}[f(\theta^N)] \leq \mathbb{E}[f(\theta^{T_\epsilon})] - \frac{\epsilon}{4} \sum_{t=T_\epsilon}^{N-1} \alpha_t \mathbb{P}(||\nabla f(\theta^t)||_2 \geq \epsilon).$$

Since $\{\mathbb{E}[f(\theta^t)]\}_{t \geq 0}$ is bounded below. Therefore:

$$\sum_{t=T_\epsilon}^{\infty} \alpha_t \mathbb{P}(||\nabla f(\theta^t)||_2 \geq \epsilon) < \infty.$$

Now define the random variables $X_t := \mathbf{1}_{\{||\nabla f(\theta^t)||_2 \geq \epsilon\}}$. We have:

$$\mathbb{E}\Big[ \sum_{t=T_\epsilon}^{\infty} \alpha_t X_t \Big] = \sum_{t=T_\epsilon}^{\infty} \alpha_t \mathbb{E}[X_t] = \sum_{t=T_\epsilon}^{\infty} \alpha_t \mathbb{P}(||\nabla f(\theta^t)||_2 \geq \epsilon) < \infty,$$

which implies $\sum_{t=T_\epsilon}^{\infty} \alpha_t X_t < \infty$ almost surely.

Since $\sum_{t=T_\epsilon}^{\infty} \alpha_t = \infty$ by assumption, $\sum_{t=T_\epsilon}^{\infty} \alpha_t X_t < \infty$ almost surely implies that $X_t = 0$ for infinitely many $t$ almost surely. In other words, for each fixed $\epsilon > 0$:

$$\mathbb{P}(\text{There are infinitely many } t \text{ such that } ||\nabla f(\theta^t)||_2 < \epsilon) = 1.$$

To establish that $\liminf_{t \to \infty} ||\nabla f(\theta^t)||_2 = 0$ almost surely, we need to show:

$$\mathbb{P}\Big( \liminf_{t \to \infty} ||\nabla f(\theta^t)||_2 = 0 \Big) = 1.$$

Note that $\liminf_{t \to \infty} ||\nabla f(\theta^t)||_2 = 0$ if and only if for every $\epsilon > 0$, there are infinitely many $t$ such that $||\nabla f(\theta^t)||_2 < \epsilon$.

Consider a sequence $\epsilon_k \downarrow 0$ (e.g., $\epsilon_k = 1/k$). For each $k$, we have:

$$\mathbb{P}(\text{There are infinitely many } t \text{ such that } ||\nabla f(\theta^t)||_2 < \epsilon_k) = 1.$$

Since this holds for each $k$ and there are countably many $k$, we can take the intersection:

$$\mathbb{P}\Big( \bigcap_{k=1}^{\infty} \{\text{there are infinitely many } t \text{ such that } ||\nabla f(\theta^t)||_2 < \epsilon_k\} \Big) = 1.$$

If a trajectory belongs to this intersection, then for every $\epsilon > 0$, we can find $k$ such that $\epsilon_k < \epsilon$, and thus there are infinitely many $t$ with $||\nabla f(\theta^t)||_2 < \epsilon_k < \epsilon$. This means $\liminf_{t \to \infty} ||\nabla f(\theta^t)||_2 = 0$ for this trajectory.

Therefore,

$$\mathbb{P}\Big( \liminf_{t \to \infty} ||\nabla f(\theta^t)||_2 = 0 \Big) = 1.$$

$\square$

