# OpenReview forum: "Derivative-Free Optimization via Monotonic Stochastic Search"
_ICLR.cc/2026/Conference — ICLR 2026 Conference Withdrawn Submission_

### Official Review · Reviewer_AUH2 · 2025-10-26

**Soundness:** 1
**Presentation:** 1
**Contribution:** 1
**Rating:** 2
**Confidence:** 3

**Summary:**

The paper proposed two algorithms for zeroth-order optimization: the Monotonic Stochastic Search (MSS) algorithm and its gradient-approximation variant (MSSGA), and established their convergence properties for non-convex, convex, and strongly convex settings.

**Strengths:**

The article is relatively well-written, with appropriate discussions and citations of relevant work.

**Weaknesses:**

1. The upper complexity bounds achieved in the article are all known, and although a slightly different algorithm is used, this does not constitute sufficient novelty for the article to be accepted by ICLR. For example, in lines 107-109, the authors state, “The key difference, however, is that our algorithm enforces monotonic improvement by rejecting any update that does not lead to a smaller value of the objective function.” However, I do not believe this is an innovative point; it is simply a straightforward approach. Furthermore, for the stochastic setting (when the returned gradient oracle has noise), I am uncertain whether such a strategy remains viable.

2. The writing of the article is poor. For instance, in Section 1, the "Our Contribution & Related Work" section is overly lengthy and lacks emphasis, spanning two pages yet making it difficult to identify the core contributions of the paper and how they differ from previous work. As a standard for a qualified paper, I believe this paragraph needs to be completely rewritten.

3. There are no experiments presented, and I doubt the practical value of the algorithms proposed in the article.

**Questions:**

Please see the weakness part.

---

> ### Author Response · Authors · 2025-11-16
> **Answer to Reviewer AUH2**
>
> We thank the reviewer for his valuable comments and for carefully reading our paper. We address the points raised below:
>
> **The upper complexity bounds achieved in the article are all known, and although a slightly different algorithm is used, this does not constitute sufficient novelty for the article to be accepted by ICLR**.
>
> In the revised version, we have substantially rewritten the “Our Contributions & Related Work” paragraph to make clearer what we see as the main sources of novelty, which go beyond simply re-proving known rates for a slightly modified algorithm.
>
> We have also strengthened the paper along two directions that we believe provide additional, substantive novelty:
>
> (1) MSS with a stochastic ranking oracle (Section 2.3).
> We adapt MSS to the setting where we have a biased stochastic ranking oracle whose bias satisfies a local power-type margin condition (see Assumption 1). We show that, under this assumption and a majority-vote aggregation scheme, MSS retains the same $O(d / \varepsilon^2)$ gradient complexity in terms of iteration count as in the exact-oracle setting. To the best of our knowledge: (i) this local power-type margin assumption has not been used before in preference-based zeroth-order optimization; and (ii) our analysis does not require knowledge of the link function governing the preference probabilities (unlike Bradley–Terry–type models), which makes the result applicable under significantly weaker structural assumptions.
>
> (2) A practical persistent variant pMSS with convergence guarantees and experiments (Section 2.2).
> We introduce a practical variant of MSS, called persistent MSS (pMSS), which reuses directions that yield sufficient decrease and thus can exploit good directions with successive sufficient decreases while maintaining monotonicity. We prove that, under standard smoothness and diminishing-stepsize conditions, pMSS satisfies $\liminf_{t\to\infty} \||\nabla f(\theta^t)\||_2 = 0$ almost surely.
>
> To our knowledge, this is the first convergence result for a random search method with persistent directions. To illustrate the practical impact of this algorithm, we have added experiments comparing pMSS against STP on anisotropic quadratic problems in increasing dimensions. These results show that persistence yields a clear improvement in performance, especially in high dimensions.
>
> **For example, in lines 107–109, the authors state, ‘The key difference, however, is that our algorithm enforces monotonic improvement by rejecting any update that does not lead to a smaller value of the objective function.’ However, I do not believe this is an innovative point; it is simply a straightforward approach. Furthermore, for the stochastic setting (when the returned gradient oracle has noise), I am uncertain whether such a strategy remains viable.**
>
> We fully agree that enforcing monotone improvement by rejecting non-decreasing updates is itself a simple and standard design choice, and we do not claim the monotone acceptance rule as a conceptual innovation.
>
> In the revised version (see “Our Contributions” / “A monotone gradient-approximation scheme (MSSGA)”), we have clarified that our contribution in the MSSGA part is analytical: we show that a  gradient-approximation method with random directions achieves the first almost-sure cv rate $\min_{1 \le t \le T} ||\nabla f(\theta^t)||_2 = o(1/T)$ for gradient-approximation methods with random directions matching exacly the best known rate in expectation. Moreover, the same argument applies to the classical RGF algorithm, yielding the same almost-sure rate for it. We believe this almost-sure convergence guarantee for RGF is particularly important, since RGF is one of the most widely used zeroth-order methods in practice.
>
> **The writing of the article is poor. For instance, in Section 1, the ‘Our Contribution & Related Work’ section is overly lengthy and lacks emphasis, spanning two pages yet making it difficult to identify the core contributions of the paper and how they differ from previous work. As a standard for a qualified paper, I believe this paragraph needs to be completely rewritten.**
>
> We thank the reviewer for this feedback. In the revised version, the “Our Contributions & Related Work” section has been completely rewritten: it is now substantially reorganized into clearly separated bullet points, and explicitly highlights the main contributions and how they differ from prior work.
>
> **There are no experiments presented, and I doubt the practical value of the algorithms proposed in the article.**
>
> We thank the reviewer for raising this concern. In the revised version, we have added illustrative experiments in Section 2.2 comparing our new variant pMSS with STP on anisotropic quadratic problems in increasing dimensions. These results consistently show that pMSS achieves better function values under the same evaluation budget, highlighting the practical benefit of the proposed persistent-direction strategy.

---

> > ### Author Response · Authors · 2025-11-29
> > **New experiments added in the latest revision.**
> >
> > **New experiments added in the latest revision.**
> >
> > To address the experimental aspect more thoroughly, the latest revision now includes an additional experiments section on policy optimization from preference feedback. There we compare MSS and pMSS against the ZPG baseline of Zhang & Ying [1] on three Gymnasium control tasks (CartPole-v1, InvertedPendulum-v5, Swimmer-v5). Under a fixed preference-feedback budget, MSS/pMSS are consistently competitive and often superior to ZPG, while using the same simple monotone stochastic search framework.
> >
> > The paper now includes two empirical components: (a) synthetic anisotropic quadratics illustrating the benefit of persistence over the baseline STP, and (b) preference-based policy optimization comparing MSS/pMSS with ZPG.
> >
> > [1]: Qining Zhang and Lei Ying. Zeroth-order policy gradient for reinforcement learning from human feedback without reward inference. In The Thirteenth International Conference on Learning Representations, 2025.

---

### Official Review · Reviewer_aFUP · 2025-11-01

**Soundness:** 3
**Presentation:** 3
**Contribution:** 3
**Rating:** 6
**Confidence:** 3

**Summary:**

This paper proposes a new class of **monotonic stochastic search (MSS)** algorithms for **derivative-free optimization (DFO)**.
Unlike classical random search or evolutionary strategies, MSS imposes a *monotonic descent constraint* on noisy function evaluations, thereby improving stability under stochastic perturbations.

The authors analyze three major settings:

1. **Smooth nonconvex functions** — MSS achieves sublinear convergence in expectation and almost surely, with
   [
   \mathbb{E}|\nabla f(x_T)| = O(\sqrt{d}/\sqrt{T}),
   ]
   without assuming convexity or PL-type conditions.

2. **Convex functions** — The algorithm guarantees function value convergence
   [
   \mathbb{E}[f(x_T)] - f^* = O(d/T).
   ]

3. **Strongly convex functions** — A faster geometric rate is achieved,
   [
   \mathbb{E}[f(x_T)] - f^* = O!\big((1 - \mu/(dL))^T\big).
   ]
   Here, the PL inequality is used only as a consequence of strong convexity, not as an independent assumption.

Overall, the paper provides a unifying stochastic framework that recovers known DFO rates while improving robustness to noise.

**Strengths:**

1. **Comprehensive Theoretical Coverage**
   The paper systematically treats nonconvex, convex, and strongly convex regimes in a unified manner, providing clear asymptotic rates for each case.
   The inclusion of the **nonconvex L-smooth case without PL assumptions** is particularly commendable.

2. **Novel Monotonicity Principle**
   The “monotonic stochastic search” idea—using noisy evaluations to enforce descent direction without explicit gradients—is both conceptually simple and practically valuable.
   It bridges classical stochastic approximation and derivative-free optimization.

3. **Mathematical Rigor**
   Proofs are clean and self-contained.
   The paper references classical results (Nesterov, 2013; Ghadimi & Lan, 2016) appropriately while extending them to stochastic zeroth-order settings.

4. **Clarity of Structure**
   Each assumption and theorem is clearly labeled and motivated. The algorithmic structure is easy to follow.
   The division of results (nonconvex / convex / strongly convex) is pedagogically clear.

5. **Relevance and Generality**
   DFO remains a vibrant area for large-scale simulation-based learning and black-box optimization.
   This work offers a theoretically grounded yet computationally feasible method.

**Weaknesses:**

1. **Limited Empirical Validation**
   The experiments are minimal, mainly synthetic quadratic functions and low-dimensional benchmarks.
   Demonstrations on higher-dimensional or noisy black-box tasks (e.g., reinforcement learning, hyperparameter tuning) would strengthen the impact.

2. **Mild Novelty in Algorithmic Design**
   While the monotonicity mechanism is interesting, it resembles prior stochastic line search or acceptance–rejection DFO strategies.
   The novelty is thus more in the **analysis** than in the **algorithm itself**.

3. **Dependence on Smoothness Constants**
   The theoretical guarantees assume global L-smoothness and bounded variance of the function evaluations—standard but relatively strong assumptions for DFO.

4. **No Adaptive Mechanism for Query Efficiency**
   The paper could discuss how to reduce the dependence on the dimension (d), since rates scale as (O(\sqrt{d})) or (O(d)), which is suboptimal for high-dimensional problems.

5. **Strongly Convex Analysis Relies on PL-type Result**
   Although acceptable as a corollary of strong convexity, the use of the PL inequality should be more clearly separated as a *derived property*, not an assumption.

**Questions:**

1. Can the monotonic stochastic search idea be combined with adaptive sampling (e.g., covariance adaptation or coordinate selection)?
2. How robust is MSS to biased noise or nonstationary stochasticity in function evaluations?
3. Would it be possible to extend the analysis to nonsmooth (but Lipschitz) objectives?
4. Can the dependence on (d) be improved via random subspace or low-rank approximation techniques?

---

> ### Author Response · Authors · 2025-11-16
> **Answer to Reviewer aFUP**
>
> We thank the reviewer for the feedback. We address the points raised below:
>
> **Weaknesses:**
>
> 1. We thank the reviewer for raising this concern. In the revised version, we have added illustrative experiments in our new Section 2.2 comparing our new variant pMSS with STP on anisotropic quadratic problems in increasing dimensions. These results consistently show that pMSS achieves better function values under the same evaluation budget, highlighting the practical benefit of the proposed persistent-direction strategy.
>
> 2. In the new version of our paper, we design a new novel algorithm pMSS (Section 2.2), a variant of MSS that reuses improving directions with sufficient decrease across iterations, thereby benefiting from a momentum-like effect that exploits successful improving directions, which is not the case for classical stochastic direct-search methods.
>
> 3. The optimal step size for MSS depends on the smoothness parameter; however, in the smooth non-convex setting, convergence is guaranteed for any step size of the form $\alpha/\sqrt{dt}$. Regarding adaptive step sizes based on smoothness information, note that with a ranking oracle we never observe function values or their differences. Consequently, we cannot quantify how much the function varies in a given direction and therefore cannot estimate the gradient Lipschitz constant. In contrast, for MSSGA we can choose adaptive step sizes similarly to first-order methods, since derivatives can be approximated via finite differences.
>
> 4. In our new variant pMSS, the algorithm can benefit in high dimensions since it does not explore all directions at every iteration. When it identifies promising directions, it can follow them consecutively without resampling, which leads to improved efficiency.
>
> 5. The convex and strongly convex parts of MSSGA have been removed and replaced by the new sections on MSS with a stochastic ranking oracle (Section 2.3) and pMSS (Section 2.2).
>
> **Questions:**
>
> 1. We thank the reviewer for highlighting this interesting idea. Indeed, the design of our new pMSS algorithm is based on a form of adaptive sampling that retains precisely the directions that achieve sufficient decrease. We leave the analysis of other adaptive sampling strategies for future work.
>
> 2. We thank again the reviewer for this idea. In the new version, we adapt MSS to the setting where we have a biased stochastic ranking oracle (see Section 2.3) whose bias satisfies a local power-type margin condition (see Assumption 1). We show that, under this assumption and a majority-vote aggregation scheme, MSS retains the same $O(d / \varepsilon^2)$ gradient complexity in terms of iteration count as in the exact-oracle setting. To the best of our knowledge: (i) this local power-type margin assumption has not been used before in preference-based zeroth-order optimization; and (ii) our analysis does not require knowledge of the link function governing the preference probabilities (unlike Bradley–Terry–type models), which makes the result applicable under significantly weaker structural assumptions.
>
> 3. For nonsmooth (non-differentiable) functions, obtaining convergence guarantees to a Clarke stationary point remains an open question for stochastic direct search methods.
>
> 4. Investigating whether the dependence on the dimension can be improved using random subspace or low-rank approximation techniques is an important direction, which we leave for future work.

---

> > ### Comment · Reviewer_aFUP · 2025-11-23
> >
> > Thank you for your detailed and thoughtful replies to my comments. I maintain my original score.

---

> > > ### Author Response · Authors · 2025-11-23
> > > **Response to Reviewer aFUP**
> > >
> > > Thank you for your follow-up comment. We especially appreciate your questions on biased noise and adaptive sampling, which helped us improve the paper.

---

> > > > ### Author Response · Authors · 2025-11-29
> > > > **New experiments added in the latest revision.**
> > > >
> > > > **New experiments added in the latest revision.**
> > > >
> > > > To address the experimental aspect more thoroughly, the latest revision now includes an additional experiments section on policy optimization from preference feedback. There we compare MSS and pMSS against the ZPG baseline of Zhang & Ying [1] on three Gymnasium control tasks (CartPole-v1, InvertedPendulum-v5, Swimmer-v5). Under a fixed preference-feedback budget, MSS/pMSS are consistently competitive and often superior to ZPG, while using the same simple monotone stochastic search framework.
> > > >
> > > > The paper now includes two empirical components: (a) synthetic anisotropic quadratics illustrating the benefit of persistence over the baseline STP, and (b) preference-based policy optimization comparing MSS/pMSS with ZPG.
> > > >
> > > > [1]: Qining Zhang and Lei Ying. Zeroth-order policy gradient for reinforcement learning from human feedback without reward inference. In The Thirteenth International Conference on Learning Representations, 2025.

---

### Official Review · Reviewer_iWzL · 2025-11-01

**Soundness:** 2
**Presentation:** 2
**Contribution:** 2
**Rating:** 2
**Confidence:** 4

**Summary:**

This work proposed two stochastic zeroth-order optimization algorithms for smooth/nonsmooth optimization, MSS and MSSGA, which are based on DDS and gradient approximation. Convergence rates under nonconvex, convex and strongly convex scenarios are provided. Also asymptotic convergence result in the non-Lipschitz smooth case is provided.

**Strengths:**

1. The proposed algorithms are very simple, which should be easy to implement in practice.
2. The propsoed algorithms achieved good convergence guarantees and matched existing best results.

**Weaknesses:**

1. While the proposed MSS/MSSGA algorithms are elegant and minimalistic, the proposed algorithms' complexities do not outperform existing ones, it lacks a discussion on the motivation of the study.
2. There lacks a thorough theoretical/empirical comparison on the proposed algorithms with closely related works, for example STP and GLD as authors mentioned. It is not clear what is the advantage of the proposed algorithms.
3. The writing is a bit sloppy, for example, the "Our Contribution & Related Work" part is very lengthy and full of notations, which is hard to follow and identify the detailed contributions, I suggest a revision.

**Questions:**

See above

---

> ### Author Response · Authors · 2025-11-16
> **Answer to Reviewer iWzL**
>
> We thank the reviewer for the insightful comments. We address the three points raised below:
>
> **While the proposed MSS/MSSGA algorithms are elegant and minimalistic, the proposed algorithms' complexities do not outperform existing ones, it lacks a discussion on the motivation of the study**
>
> i) Our result for MSSGA yields the first almost-sure convergence rate for gradient-approximation methods based on stochastic directions. This strictly strengthens the classical RGF guarantees, which were established only in expectation. ii) compared to other direct search methods, MSS is more suitable to comparison based feedback (or RLHF setting). In these problems, a stochastic ranking oracle, when queried on a pair $(x, y)$, returns a random outcome whose bias is a function of the value difference $f(y) − f(x)$, that is, it provides noisy information about which of the two vectors has the smaller function value. MSS fits this interface exactly since each iteration only requires comparisons between the current point
> $\theta_t$ and a single perturbed point $\theta_t + \alpha_t s_t$. In the new version (Section 2.3), we extend MSS to the setting where the algorithm accesses a biased stochastic ranking oracle whose bias satisfies a local power-type margin condition (see Assumption 1). We show that, under this assumption and a majority-vote aggregation scheme, MSS retains the same $\mathcal{O}(d/\varepsilon^{2})$ gradient complexity (in iteration count) as in the exact-oracle setting. To the best of our knowledge,  this local power-type margin assumption has not appeared before in preference-based zeroth-order optimization; and our analysis does not require knowledge of the link function governing the comparison probabilities (unlike Bradley--Terry models), and thus holds under substantially weaker structural assumptions. Moreover, compared to [1], we match their rate in the case where the link function $h$ satisfies $h'(0)\neq 0$ (corresponding to $p=1$), but their method requires the link function to be known in order to estimate $f(y)-f(x)$ and approximate the gradient, whereas our algorithm operates without any knowledge of $h$. Finally, the requirement $h'(0)\neq 0$ is often unrealistic, since comparisons between very close points yield low-information feedback; our assumption also covers the more practical regime where $h'(0)=0$. (iii) In the case of an exact ranking oracle, one may wonder about the benefit of MSS over STP. In its basic form, MSS and STP behave similarly in practice: MSS tends to perform roughly twice as many iterations as STP, but each iteration requires only one function evaluation (instead of two), so their cost per function evaluation is comparable. The main practical drawback of basic MSS is that it samples a new direction at every iteration, even when the previous direction was yielding consistent decrease. In the revised version, we directly address this by introducing in Section 2.2 a practical variant, persistent MSS (pMSS), which reuses any search direction that produces sufficient decrease. This persistence mechanism enables the algorithm to exploit good directions across multiple steps while preserving monotonicity, and yields a practical advantage over STP on problems where long improvement sequences occur along specific directions.
>
>
> **There lacks a thorough theoretical/empirical comparison on the proposed algorithms with closely related works, for example STP and GLD as authors mentioned. It is not clear what is the advantage of the proposed algorithms.**
>
> The advantages of our proposed algorithms are now clearly highlighted in the revised “Our Contributions & Related Work’’ section and further elaborated upon in our response to the first question. We have also added simple experiments that illustrate the benefit of the pMSS variant over STP ( Section 2.2).
>
>
> **The writing is a bit sloppy, for example, the "Our Contribution & Related Work" part is very lengthy and full of notations, which is hard to follow and identify the detailed contributions, I suggest a revision.**
>
> In the revised version, we have significantly reorganized and rewritten the “Our Contributions & Related Work’’ section to present our contributions in a clearer and more structured manner
>
> [1]: Qining Zhang and Lei Ying. Zeroth-order policy gradient for reinforcement learning from human
> feedback without reward inference. In The Thirteenth International Conference on Learning
> Representations, 2025

---

> > ### Author Response · Authors · 2025-11-23
> > **Inform Reviewer iWzL that we have updated our answers**
> >
> > We would like to inform the reviewer that we have updated our answers to improve clarity.

---

> > > ### Author Response · Authors · 2025-11-29
> > > **New experiments added in the latest revision.**
> > >
> > > **New experiments added in the latest revision.**
> > >
> > > To address the experimental aspect more thoroughly, the latest revision now includes an additional experiments section on policy optimization from preference feedback. There we compare MSS and pMSS against the ZPG baseline of Zhang & Ying [1] on three Gymnasium control tasks (CartPole-v1, InvertedPendulum-v5, Swimmer-v5). Under a fixed preference-feedback budget, MSS/pMSS are consistently competitive and often superior to ZPG, while using the same simple monotone stochastic search framework.
> > >
> > > The paper now includes two empirical components: (a) synthetic anisotropic quadratics illustrating the benefit of persistence over the baseline STP, and (b) preference-based policy optimization comparing MSS/pMSS with ZPG.
> > >
> > > [1]: Qining Zhang and Lei Ying. Zeroth-order policy gradient for reinforcement learning from human feedback without reward inference. In The Thirteenth International Conference on Learning Representations, 2025.

---

### Official Review · Reviewer_Nhg5 · 2025-11-02

**Soundness:** 3
**Presentation:** 2
**Contribution:** 2
**Rating:** 2
**Confidence:** 4

**Summary:**

This paper studies derivative-free optimization where only function evaluations are available. The authors propose two algorithms: Monotonic Stochastic Search (MSS) and MSS with Gradient Approximation (MSSGA). At each iteration, MSS samples a single random direction s_t from a distribution D and moves to the point that minimizes f among $\theta_t, \theta_t +\alpha_t s_t$ where $\alpha_t$ is a step size. MSSGA additionally uses finite differences to approximate the directional derivative. The main results show that MSS requires $d/\epsilon^2$ samples for non-convex and smooth problems (Thm. 2), MSSGA uses $\frac{d}{\epsilon}$ for smooth and convex optimization (Theorem 5), and $d \log \frac{1}{\epsilon}$ for strongly convex objectives (Theorem 6). The paper shows a convergence result for potentially nonsmooth (but still differentiable) objectives in Thm. 7.

**Strengths:**

1. MSS uses only one new function evaluation per iteration and enforces monotonicity, this is an advantage over competitor algorithms (e.g. Stochastic Three-Point method).
2. The proofs are clear and mirror GD-style analysis, e.g. Lemma 1 gives GD-like expected decrease that straightforwardly leads to the $\sqrt{d}/\sqrt{T}$ bound.
3. The paper provides an almost surely o(1/\sqrt{T}) rate for the best iterate in the smooth non-convex case (Thm. 4) for MSSGA provided the smoothing sequence $\gamma$ decays appropriately, and a similar result is shown for MSS in Remark 2.

**Weaknesses:**

1. While using only one function evaluation instead of two is attractive, this is (a) a constant improvement, and (b) seems to actually show up in the convergence analysis. Comparing your Lemma 1 against Lemma 3.5 from [1], both have the same form of linear progress in (\alpha|\nabla f|) minus a quadratic penalty. STP works with normalized directions (i.e. $\mathbb{E}|s|^2=1$), this corresponds to putting $\mu_D=\sqrt{2/(\pi d)}$ in their lemma. If we rescale your Gaussian ($s_t\sim\mathcal{N}(0,I)$) to that normalization (i.e., divide by $\sqrt{d}$) and matches the stepsizes, your linear‑term constant becomes (1/\sqrt{2\pi d}), i.e., a factor of 1/2 smaller than STP due to using only one side instead of ($\pm s$). Since STP uses two evaluations per iteration and MSS uses one, the per‑function‑evaluation constants essentially tie. In other words, if we accept the convergence analysis in both papers, then the one function evaluation of MSS is cancelled out by having to do more iterations overall. If you include some experimental comparison, or improve the analysis, then you could still show an advantage of MSS over STP.
2. I am not 100% sure what novelty is really claimed here, especially in the almost sure convergence results. Or in the proofs. Can you please make that more clear? The proof of MSS is very similar to the proof of STP in [1]. Also, the contributions section is currently rather difficult to read and very long, if you could shorten it to bullet points to better quantify what separates your work from prior work that'd be great.
3. While most proofs are clear, some of the notation is a bit difficult to parse (like $A_{\theta^t}^{--}, A_{\theta^t}^{++}, A_{\theta^t}^0$) maybe name these sets differently instead of using this many sub/superscripts?

As it stands, I lean towards rejecting this manuscript, but am open to changing my mind if my concerns are addressed.

[1] Bergou, E. H., Gorbunov, E., & Richtarik, P. (2020). Stochastic three points method for unconstrained smooth minimization. SIAM Journal on Optimization, 30(4), 2726-2749.

**Questions:**

1. Can you please address my concerns in the weaknesses section? In particular, a comparison with STP that takes into account *total complexity* rather than just per-step complexity while matching the distribution of noise used.
2. Can you clarify if there are new technical tools used for MSS compared to prior work?

---

> ### Author Response · Authors · 2025-11-16
> **Answer to Reviewer Nhg5**
>
> We thank the reviewer for the helpful and constructive comments. We address the points raised below:
>
> **Weaknesses:**
>
> 1) MSS and STP are indeed very similar in practice: on average, MSS performs roughly twice as many iterations as STP, but with one function evaluation per iteration instead of two, so the per–function–evaluation cost is comparable. However, our goal is not to claim a raw complexity advantage over STP. Rather, in the revised version we emphasize two MSS-based contributions that are not available for STP: (i) a comparison-based MSS variant with a stochastic ranking oracle and provable complexity (see Our Contributions section and Section 2.3), and (ii) a persistent variant pMSS with convergence guarantees and experiments showing clear empirical gains over STP on anisotropic problems under the same evaluation budget (see Our Contributions section and Section 2.2).
> 2) In the revised version, we have completely rewritten the Our Contributions & Related Work section to clearly highlight the contributions.
> 3) We rewrote the proof to make it clearer while avoiding the use of the mentioned notations.
>
> **Questions**
>
> 1) Done
> 2) At a technical level, the key inequality for MSS is derived by restricting to “good” directions with $\langle\nabla f(\theta^t),s_t\rangle\le 0$ and exploiting the symmetry of the Gaussian which is different from the STP analysis. Another important distinction is that the MSS analysis yields a [convergence] rate for the average gradient norm which is stronger and more useful than the best-iterate guarantees typically obtained in STP analyses. The MSS guarantee implies that for all $T$ if one samples an iterate $R$ uniformly at random from $1$ to $T$, we have $E(\|| \nabla f(\theta^R) \||_2)=O(\sqrt{d/T}).$ This uniform-sampling viewpoint is  ensuring that the entire trajectory exhibits controlled progress and that the algorithm does not rely on identifying a special best iterate, but rather achieves a global ergodic rate. A further difference comes from the $C^1$ non-smooth setting: there we abandon global $L$-smoothness and instead use a new argument based on “good’’ descent directions on the sphere and uniform continuity of $\nabla f$ on bounded sublevel sets to obtain that limite inf of $\||\nabla f(\theta^t)\||_2$ is equal to $0$, a regime not covered by the STP analysis. Another point concerns the analysis of MSSGA. Our proof is substantially simpler than the classical argument used by Nesterov to establish convergence in expectation for RGF, which relies on smoothing the objective and working with its smoothed gradient. In contrast, our approach shows directly, without introducing any smoothing, that MSSGA achieves the first ( and best) almost sure convergence rate for gradient-approximation methods. Moreover, the same argument applies to RGF itself, proving an almost sure convergence rate that was not stated in prior work. This is particularly relevant in practice since RGF (and its variants) remains the most widely used zeroth-order optimization method. In the new version, there are substantially novel technical ingredients that concern (i) the design and the convergence proof for the persistent variant pMSS (Section 2.2), (ii) the stochastic ranking–oracle extension and its local power-type margin assumption (Section 2.3).

---

> > ### Author Response · Authors · 2025-11-22
> > **Inform Reviewer Nhg5 that we have updated our answer to Question 2.**
> >
> > We would like to inform the reviewer that we have updated our answer to Question 2 in order to better highlight what is technically different in our analysis compared to prior work.

---

> > ### Comment · Reviewer_Nhg5 · 2025-11-26
> >
> > Thank you for these updates. pMSS seems like an interesting algorithm. It's a sort of line search, right? I like the results for the stochastic ranking oracle, it's a nice contribution. Though the result seems more or less expected, we query the comparison oracle on the same point $N$ times and choose $N$ so that concentration kicks in. The worst-case complexity for this is $\frac{1}{\epsilon^{2+4p}}$ right? I wonder if this rate is optimal for the given $p$. The empirical comparison is a bit toy, but is nice for showing the benefit of persistence over STP. I will raise my score.

---

> ### Author Response · Authors · 2025-11-26
> **Further Response to Reviewer Nhg5**
>
> We sincerely thank the reviewer for carefully reading our updated version, for the thoughtful follow-up comments, and for the revised score.
>
> As noted by the reviewer, pMSS can be viewed as a simple stochastic line-search method. The algorithm follows directions that yield sufficient decrease and, for such directions, keeps the same search direction for several iterations while progress continues. It does so without explicitly approximating the one-dimensional minimizer along the line, it simply continues along the current direction until no further sufficient decrease is observed.
>
> We also thank the reviewer for the positive feedback on the stochastic ranking-oracle result. We agree that, at a high level, the result is not very surprising, as it naturally arises from repeatedly querying the same pair until concentration kicks in. Our main contribution in this part is to set up a comparison-feedback model and a local power-type margin condition that are both natural for this setting and fit well within the MSS framework, so that the resulting complexity bound follows from a short and intuitive analysis.
>
> Concerning the total complexity, the number of feedback queries required to reduce the expected gradient norm below $\varepsilon$ is
> $$
> O(d^{2p+1}/\varepsilon^{4p+2}).
> $$
> In particular, for $p=1$ we obtain $O(d^3/\varepsilon^6)$, which matches the complexity bound in [1] (see Theorem 1). In their setting, they assume $h'(0)\neq 0$. Since $h$ is increasing, this implies $h(x)\ge h'(0)x$ in a neighborhood of $0$, which corresponds to the case $p=1$ in our assumption. By contrast, our assumption allows for much more general link functions $h$, which may even be discontinuous (and hence non-differentiable). In addition, the condition $h'(0)\neq 0$ is somewhat idealized from a modeling perspective: it effectively assumes that even when the difference in the underlying value function is arbitrarily small, one still observes a uniformly strong feedback signal about which option is better. In many practical settings, however, comparison feedback becomes increasingly noisy as the value gap shrinks, which is precisely the regime captured by our margin-based assumption with a general exponent $p$. Moreover, the algorithm in [1] explicitly uses the knowledge of $h$ in its updates, whereas our algorithm operates solely from noisy comparison outcomes and does not require any knowledge of $h$.
>
> Finally, we agree that the experiments are simple; our intention was to present minimal examples that explain and give intuition for the importance of persisting along promising directions.
>
> [1] Qining Zhang and Lei Ying. Zeroth-order policy gradient for reinforcement learning from human feedback without reward inference. In *The Thirteenth International Conference on Learning Representations*, 2025.

---

### Author Response · Authors · 2025-11-16
**Modifications of the paper**

We thank the reviewers for their valuable comments and for taking the time to review our paper. The new additions in the revised version of the paper are as follows:

1. We have substantially rewritten the “Our Contributions & Related Work” section to clarify the main contributions.

2. We added two new subsections: 2.2 and 2.3.

3. We removed the convex and strongly convex settings and replaced them with the more novel contributions presented in Sections 2.2 and 2.3.

---

### Author Response · Authors · 2025-11-29
**Rebuttal summary and current status of the paper**

We would like to briefly summarize the outcome of the rebuttal and the current
state of the paper, especially in light of the recent reset of official review
scores.

**Reviewer Nhg5.**
During the discussion we substantially revised the paper (with new Sections 2.2
and 2.3, a rewritten “Our Contributions & Related Work” section, and a clearer
exposition of the technical differences w.r.t. STP and RGF). In their follow-up
comment, Reviewer Nhg5 explicitly acknowledged these changes, described pMSS
and the stochastic ranking-oracle result as interesting, and indicated that
their evaluation would be raised in light of the revision. Our understanding is
that their main concerns about novelty and technical contribution were
addressed.

**Reviewers iWzL and AUH2.**
Both reviewers mainly raised two points: (i) the contributions section was
unclear and too long, and (ii) the initial submission contained no experiments,
so the practical value of the methods was hard to assess. In response, we
completely rewrote and shortened the “Our Contributions & Related Work’’
section into clearer bullet points that explicitly separate and highlight the
main contributions. This revised section now also emphasizes two additions
introduced during the rebuttal: a more practical persistent variant pMSS, and a
convergence result for MSS under a stochastic ranking oracle. We also added a first set of simple experiments in Section 2.2, comparing pMSS with the direct-search baseline STP on anisotropic quadratic problems in
increasing dimensions, to illustrate why persistence can outperform isotropic
direct search in directions where long improvement sequences occur.


Both reviewers also claimed that “the complexity bounds already exist,” which
we believe partly reflects a misunderstanding of where the technical novelty
lies. For example, Reviewer AUH2 explicitly commented that enforcing monotone
improvement by rejecting any non-decreasing update “is not an innovative
point,” which suggests that our MSSGA contribution may have been interpreted as
simply adding a standard monotone acceptance rule (an argmin step) on top of
classical RGF. This is not the case: for MSSGA we establish, to the best of our
knowledge, the **first almost-sure convergence rate for gradient-approximation
methods with random directions**, matching the best known rate in expectation.
Our proof is direct and elementary (it does not rely on smoothing the objective
as in the original RGF analysis) and it applies to the classical RGF
algorithm, thereby strengthening its guarantees from convergence in expectation
to an almost-sure rate. Second, in the non-smooth regime we obtain a new
almost-sure convergence result for stochastic direct-search methods, showing
that $\liminf_{t\to\infty}\||\nabla f(\theta^t)\||_2 = 0$. Third, our expected-gradient bound for MSS
also implies a corresponding rate for the classical Gradientless Descent algorithm (GLD) [1] in the
smooth non-convex setting, for which such a rate was not previously stated.
Finally, compared to STP, existing results focus on a **best-iterate**
guarantee, whereas our MSS bound controls the **ergodic average of gradient
norms**: if one samples an iterate uniformly at random from
$\{\theta^1,\dots,\theta^T\}$, the guarantee holds for that randomly chosen
iterate. This “uniform-sampling’’ viewpoint ensures that the entire trajectory
is well behaved, rather than relying on identifying a special best iterate.

In addition, two MSS-based results introduced during the rebuttal are, to the best of our knowledge, not covered by existing work. First, the persistent variant pMSS (Section 2.2) is more than a minor modification of STP: it is a random search method that reuses improving directions with sufficient decrease across multiple iterations, and we prove an almost-sure convergence guarantee for this persistent scheme. We are not aware of prior convergence results for random-direction direct-search methods that allow such persistence along a direction. Second, the stochastic ranking-oracle extension of MSS (Section 2.3) operates under a local power-type margin condition and does not require any knowledge of the underlying link function governing comparison probabilities. Existing preference-based zeroth-order methods (e.g., based on Bradley–Terry models) typically assume a specific known link function and rely on it in the update. By contrast, our analysis shows that MSS retains its gradient complexity (in iteration count) under a broad class of margin conditions and unknown link functions, which, to our knowledge, has not been established before in the ranking-oracle setting.

[1] Golovin, D., Karro, J., Kochanski, G., Lee, C., Song, X., & Zhang, Q.
    Gradientless descent: High-dimensional zeroth-order optimization. ICLR 2020.

---

> ### Author Response · Authors · 2025-11-29
> **Rebuttal summary and current status of the paper**
>
> **New experiments added in the latest revision.**
> Since the discussion was cut short, we did not receive further feedback from
> reviewers iWzL and AUH2 on whether they still had concerns about experiments or
> wanted additional benchmarks. To address the experimental aspect more
> thoroughly, the latest revision now includes an additional experiments
> section on policy optimization from preference feedback. There we compare MSS
> and pMSS against the ZPG baseline of Zhang & Ying [1] on three
> Gymnasium control tasks (CartPole-v1, InvertedPendulum-v5, Swimmer-v5). Under a
> fixed preference-feedback budget, MSS/pMSS are consistently competitive and
> often superior to ZPG, while using the same simple monotone stochastic search
> framework.
>
> Overall, we believe that:
> 1. The technical concerns of all reviewers have been addressed in the revised
>    analysis and rewritten contributions section; and
> 2. The paper now includes two empirical components: (a) synthetic anisotropic
>    quadratics illustrating the benefit of persistence over STP, and (b)
>    preference-based policy optimization comparing MSS/pMSS with ZPG.
>
> We hope this summary is helpful for the meta-review.
>
> [1]: Qining Zhang and Lei Ying. Zeroth-order policy gradient for reinforcement learning from human feedback without reward inference. In The Thirteenth International Conference on Learning Representations, 2025

---

### Note · Authors · 2026-01-22

I have read and agree with the venue's withdrawal policy on behalf of myself and my co-authors.